# High-spin state dynamics and quintet-mediated emission in intramolecular singlet fission

Jeannine Grüne [1,2,6] ✉, Steph Montanaro[3], Thomas W. Bradbury [4], Ashish Sharma[1], Simon Dowland[1], Alexander J. Gillett [1], Sebastian Gorgon [1], Oliver Millington [3], William K. Myers [2], Jan Behrends [5], Jenny Clark [4], Akshay Rao [1], Hugo Bronstein [1,3] & Neil C. Greenham [1] ✉

High-spin states in molecular systems hold significant interest for applications ranging from optoelectronics to quantum technologies. Spin states generated in intramolecular singlet fission are of particular relevance, yet the mechanisms controlling triplet-pair formation are not fully understood – especially the involvement of quintet states in luminescence at room temperature remains experimentally elusive. Here, we investigate high-spin state formation and emission in dimers and trimers comprising multiple diphenylhexatriene units. We demonstrate the formation of pure quintet states in all these oligomers, with quintet-mediated emission dominating delayed fluorescence up to room temperature. By distinguishing between the formation of weakly exchange-coupled triplet pairs and triplet excitons generated by intersystem crossing, we identify the methylated trimer as the only oligomer exhibiting exclusively the desired singlet fission route. These findings establish quintet-mediated delayed emission as a distinct spin-selective pathway and show how molecular structure directs high-spin formation, opening opportunities for room-temperature molecular quantum technologies.

Molecular high-spin states present a diverse platform for organic optoelectronics and quantum technologies that involve complex processes requiring transition between different spin states[1–4]. One key application is singlet fission (SF), a fast, spin-conserving photophysical process converting singlet excitons rapidly into spin-correlated triplet pairs, before these dissociate into free triplets[1,5]. This principle has a significant potential impact on enhancing photovoltaic efficiency as photocurrent generation occurs from each triplet state[6,7]. Additionally, the formation of quintet states emerges as a critical aspect of the fission process, offering potential applications beyond photovoltaics, such as in quantum computing and spintronics[8,9].

Whilst intermolecular SF has been extensively studied, we focus on intramolecular singlet fission (iSF), where molecular structures can be tuned to control the rates and generation of triplet pairs[10–15]. Recent literature has primarily explored iSF in dimers, for instance of pentacene[16,17] and tetracene[18–20], with only a few recent reports extending to trimer studies, which often exhibit distinct SF behaviour[21,22]. While these studies have provided insight into structural control and triplet dissociation, key questions remain regarding the formation and emissive pathways of high-spin states: (1) How do chromophore number and arrangement affect triplet pair separation and suppression of competing intersystem crossing (ISC)? (2) Do the high-spin states, particular the quintet states, participate in emission?

[1]Cavendish Laboratory, University of Cambridge, Cambridge, UK. [2]Centre for Advanced Electron Spin Resonance, University of Oxford, Oxford, UK. [3]Department of Chemistry, University of Cambridge, Cambridge, UK. [4]Department of Physics and Astronomy, University of Sheffield, Sheffield, UK. [5]Fachbereich Physik, Freie Universität Berlin, Berlin, Germany. [6]Present address: Department of Materials, University of Oxford, Oxford, UK. ✉e-mail: jeannine.grune@materials.ox.ac.uk; ncg11@cam.ac.uk

Previous studies have shown the importance of quintet formation and their spin dynamics in singlet fission systems, for instance in crystals, tetracene oligomers[23] and metal-organic frameworks[8], where conformational modulation of exchange interactions was linked to triplet pair dissociation and spin coherence. However, these works did not report optical spin readout, notably, room-temperature emission mediated by quintet states has not been addressed. This leaves a critical gap in understanding how high-spin states function as radiative intermediates and how they could be harnessed for spin-based technologies.

The magnetic field effect (MFE) on photoluminescence (PL) is a valuable technique to reveal spin-sensitive emission behaviours and even exchange interaction at higher fields[24]. However, it relies on passive field-induced perturbation of spin levels and is unable to distinguish the exact states involved. Optically detected magnetic resonance (ODMR) allows direct microwave-driven manipulation of spin sublevels, enabling a precise optical readout of the participating high-spin states. ODMR studies of SF materials have so far been limited to low temperatures or crystalline samples. To our knowledge, no report has demonstrated room-temperature ODMR detection of quintet-mediated emission in iSF systems or generally in molecular systems.

Here, we investigate diphenylhexatriene (DPH)-based oligomers to identify structural factors that govern quintet formation and reveal their contribution to PL. Next to a directly linked dimer (DPH)$_2$, and trimer (DPH)$_3$, we introduce a methylated trimer Me-(DPH)$_3$ in this study. We focus particularly on the role of quintet-mediated delayed emission, as well as aiming to discriminate between formation of weakly-coupled triplet pairs and the formation of triplet excitons by ISC. Our previous work showed that these materials support triplet pair generation[25], but the detailed spin-state formation and especially emission pathways, addressable by magnetic resonance techniques, remained unrevealed. By introducing methyl substitution to control molecular planarity[26], we examine how structure influences the formation and emission of quintet and triplet states. Our goal is to identify molecules that can effectively emit via quintet states up to room

temperatures, as well as suppress efficiency-limiting pathways as ISC, to optimize materials for both energy conversion and quantum technology applications.

We address these questions with a comprehensive set of advanced spin-sensitive optical and magnetic resonance techniques, combining transient electron paramagnetic resonance (trEPR), ODMR, MFE, and transient absorption (TA) to map the formation and emission of states with different spin multiplicities. The unique complementarity of these techniques is further discussed in Supplementary Note 3. In contrast to previous studies, this approach directly resolves the generation of quintet and triplet-pair states and provides clear experimental evidence for quintet-mediated emission.

In this study, we show that quintet states are generated across all studied oligomers and provide an optical readout pathway via ODMR up to room temperature, confirming their emissive character and long-lived nature in delayed fluorescence. Moreover, we distinguish between triplet multiplication via SF and triplet generation via ISC, and show that Me-(DPH)$_3$ predominantly exhibits SF-derived triplet formation above 80 K, with ISC contributions emerging only at lower temperatures. These findings establish DPH-based oligomers as model systems for understanding and exploiting quintet-mediated emission in high-spin molecular materials relevant to energy conversion and quantum photonics.

## Results
### Probing high-spin state formation by electron paramagnetic resonance

Figure 1a illustrates the molecular structures and corresponding trEPR measurements for the three oligomers studied to investigate the formation of triplet states and higher spin multiplicity pair states. As in our previous reports, (DPH)$_2$ and (DPH)$_3$ consist of two or three directly connected DPH chromophores, with terminal 2'-ethylhexyl groups (DEH) to aid solubility[25]. The trimer Me-(DPH)$_3$ introduced here is distinguished by the inner unit being replaced by 1,6-dimethyl-diphenylhexatriene to study the impact of methyl substituents, as

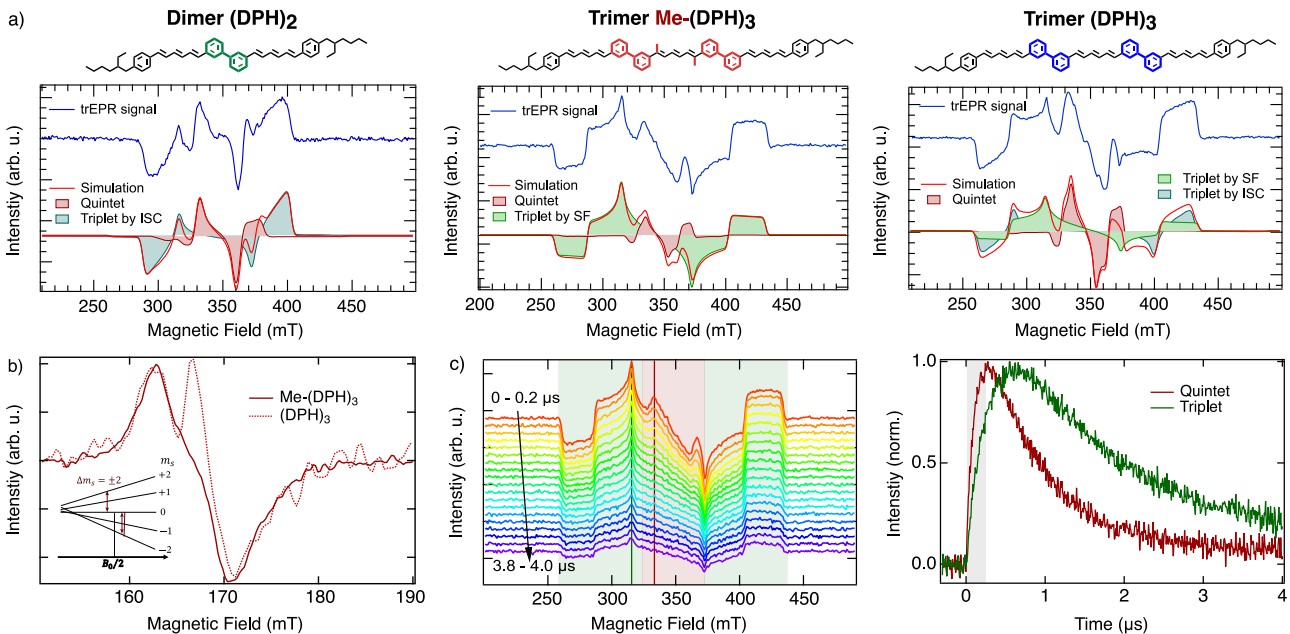

**Fig. 1 | Electron paramagnetic resonance to probe quintet and triplet pair generation. a** Molecular structures and trEPR spectra (blue) including simulation (red) from 0 to 500 ns for (DPH)$_2$, Me-(DPH)$_3$ and (DPH)$_3$. Me-(DPH)$_3$ reveals quintet (red filled) formation and weakly-coupled triplet pairs (green filled) via singlet fission process, whilst (DPH)$_2$ and (DPH)$_3$ show triplet formation only or additionally via ISC (blue filled). **b** Half-field (HF) signal ($\Delta m_s = 2$ transitions) reveals only quintet features for Me-(DPH)$_3$ whilst (DPH)$_3$ additionally exhibit HF signal from ISC triplets. **c** TrEPR spectra from 0 to 4 μs and extracted kinetics for the quintet ($B = 335$ mT, red) and triplet ($B = 305$ mT, green), that show long-lived quintet and triplet features with different peak times and decay behaviour (response time marked in grey). Measurements performed at $\lambda = 355$ nm and $T = 80$ K.

previously used in monomers and already established in carotenoids (for further synthetic details, see Supporting Information)[1,27,28]. Optical characterization confirms that Me-$(DPH)_3$ retains comparable electronic properties to the previous system $(DPH)_3$, supporting the use of these oligomers as comparable models to investigate high-spin state behaviour. Slight changes in net absorption of the $S_0$–$S_2$ electronic states are visible upon methylation, which is commonly seen in carotenoids where methyl-induced torsion slightly reduces effective conjugation and electronic delocalization[29] (further discussed in Supplementary Note 4).

The trEPR spectra of $(DPH)_2$, Me-$(DPH)_3$, and $(DPH)_3$ are shown averaged from 0 to 500 ns (blue). All spectra exhibit multiple transitions, indicating non-thermalized high-spin states. Common features in all spectra include absorptive/emissive transitions at 330 mT and 360 mT, respectively, with variations in outer transitions. Simulations (red, EasySpin[30] parameters in Supplementary Table 1) and pulsed EPR measurements (Supplementary Fig. 1) identify the inner transitions as the $m_s = 0 \rightarrow \pm 1$ transition of the strongly correlated triplet pair $^5$TT, representing a pure quintet state with $S = 2$[5,14]. The zero-field splitting parameters $|D_Q| = 750$ MHz and $|E_Q| = 80$ MHz are consistent with the expected splitting of $|D_T|/3$ ($|D_T| = 2450$ MHz as shown later) for a coupled triplet pair $^5$TT in the strong exchange limit[5,31]. The lifetime of the quintet state (Fig. 1c) as well as the ability to measure $^5$TT in ODMR (see below) supports the appearance of a longer-lived pure quintet state within the strongly-coupled regime.

Several models have been proposed to describe how quintet state $^5$TT emerges from an initially formed $S = 0$ triplet pair $^1$TT. In the low-$J$ regime, where the exchange interaction $J(t)$ transiently becomes small and comparable to the zero-field splitting $D$, coherent singlet-quintet mixing can occur on nanosecond timescales[5,16]. A subsequent increase in $J$ projects the mixed state onto $^5$TT, resulting in spin-polarized quintets. In contrast, another mechanism discussed in the literature includes $J$ remaining large but dynamically fluctuating due to structural or vibronic motions, which can enable anisotropic spin-relaxation pathways into the quintet manifold[32,33]. These include spin polarization transfer from $^1$TT or $^5$TT to weakly coupled T + T states, modulated by torsional conformations and dynamic $J$-coupling[33]. As we discuss below, our experimental data on the dimer suggest the latter picture—where $J$ mostly remains high but fluctuates dynamically, briefly allowing spin-relaxation into $^5$TT without requiring persistent occupation of the weakly-coupled regime.

Whilst quintet state formation is verified in all three oligomers, triplet state formation differs, as represented in the additional outer transitions. A reduction of effective $J$ can bring the system into a weakly-coupled triplet-pair manifold $^n$(T...T) $\leftrightarrow$ $T_1 + T_1$, in which the two triplets increase separation whilst preserving spin coherence. Thus, they inherit the selective $m_s = 0$ level population, resulting in an *eaaeea* pattern (with microwave emission *e* and absorption *a* in the three canonical orientations, see explanation in Supplementary Note 6) with the $D$ value of the triplet. Reviewing Me-$(DPH)_3$, the outer transitions (shaded green) can be attributed to this pathway, evidenced by the selective $m_s = 0$ population. The ZFS parameters $|D_T| = 2450$ MHz and $|E_T| = 270$ MHz match with those of the DEH-DPH monomer (Supplementary Fig. 2), suggesting the SF-derived triplet pairs extending to the outer chromophores.

In contrast, the triplet features in $(DPH)_2$ and $(DPH)_3$ cannot be solely reproduced by SF-derived triplet polarization. For the dimer $(DPH)_2$, the observed triplet signal can be attributed to ISC origin, leading to a population of the zero-field sublevels of $[p_x, p_y, p_z] = [0\ 1.00\ 0.52]$ with $|D_T| = 2450$ MHz. This assignment is supported by measurements on DEH-DPH monomers in dilute toluene solution, that can only undergo ISC and display the same pattern in trEPR (Supplementary Fig. 2). Thus, the dimer is effectively unable to access weakly coupled triplet states, as also indicated by the absence of a MFE (Supplementary Fig. 3), while still producing strongly coupled quintet

states. This suggests that effective dissociation into weakly exchange-coupled triplet pairs is suppressed with two directly linked DPH chromophores. Instead, $^5$TT formation is enabled by subtle structural fluctuations that transiently modulate the exchange interaction $J$ within a strongly coupled triplet pair, allowing spin evolution from $^1$TT to $^5$TT. This mechanism is consistent with theoretical predictions and was recently shown in acene dimers and macrocyclic pentacene dimers[34–36]. The observation of quintet states confirms that even the dimer undergoes fast initial singlet fission to $^1$TT, as also verified by TA measurements (see below). However, by establishing an equilibrium between $^n$TT and emitting singlet states, an ISC pathway to $T_1$ ultimately occurs. Notably, this ISC feature does not appear in trEPR of Me-$(DPH)_3$ at 80 K, excluding this ISC signal from being a relevant pathway in the formation of triplet pair states.

Strikingly, $(DPH)_3$ exhibits a distinct spectral shape that does not correspond to SF or ISC mechanisms alone. However, by combining the simulation from Me-$(DPH)_3$ with the ISC pattern from $(DPH)_2$, the spectra can be accurately reproduced. While the SF-derived weakly-coupled triplet pairs in (Me)-$(DPH)_3$, and the ISC-generated triplets in $(DPH)_2$ and in the monomer exhibit a $|D_T|$ value of 2450 MHz, the ISC-generated triplet in $(DPH)_3$ shows an increased $|D_{T,\ ISC}|$ value of 3500 MHz with $[p_x, p_y, p_z] = [0\ 1.00\ 0.52]$, indicating an increased localization within the trimer given the more complex energetic landscape[37].

To selectively probe the quintet state, we measured the transitions at half the resonant magnetic field $B_0$, known as the half-field (HF) signal (Fig. 1b)[38]. Me-$(DPH)_3$ exhibits a derivative-shaped HF signal with both absorption and emission features, which can only stem from a quintet state with multiple possible transitions (see inset). Examining $(DPH)_3$, we observe the same HF signal as the quintet state, along with an additional triplet HF signal. In contrast to SF-generated triplet states with selective $m_s = 0$ population, ISC-derived triplets in $(DPH)_3$ have sufficient spin polarization between $m_s = \pm 1$ to generate a HF signal, confirming the observation from the trEPR spectra in full-field.

To better understand temporal spin state formation, we examined the time evolution of the EPR signal. Figure 1c displays the trEPR spectra for Me-$(DPH)_3$ in 200 ns intervals up to 4 μs. Initially, the quintet signal (red shaded) dominates, peaking within the instrument response time (100 ns, grey shaded) and decays within a time of 2 μs. In contrast, the triplet signal (green) peaks later at 800 ns and displays a more prolonged decay. These timescales show that both states are long-living with the faster decay of the quintet state attributable to a pathway leading to measurable PL and providing optical readout, which will be discussed below.

## Revealing quintet-mediated emission by spin-sensitive PL

Whilst EPR shows the possibility of manipulation of the high-spin states, readout can be performed by optical techniques, such as ODMR[9,39]. ODMR employs the principles of EPR by using microwave irradiation to alter the spin polarization between sublevels. Under continuous illumination, simulating realistic conditions, the PL readout changes as the sublevel population varies under magnetic resonance[40].

Figure 2a presents the ODMR spectra (blue) including simulations (red) for the dimer $(DPH)_2$ and both trimers, Me-$(DPH)_3$ and $(DPH)_3$, at $T = 80$ K (see Supplementary Table 3 for simulation parameters), whilst the temperature dependence from 10 K to 293 K is discussed further below. The dimer displays only the response of the quintet state, whilst the ISC triplet does not participate in the PL, proving an optical pathway from the quintet state via the reverse SF pathway to the emissive singlet state (discussed below). The quintet state is also visible in both trimers, Me-$(DPH)_3$ and $(DPH)_3$, with the triplet pairs being visible as well at $T = 80$ K. The spectral features are consistent with the EPR simulation in Fig. 1, confirming the assignment of quintet and

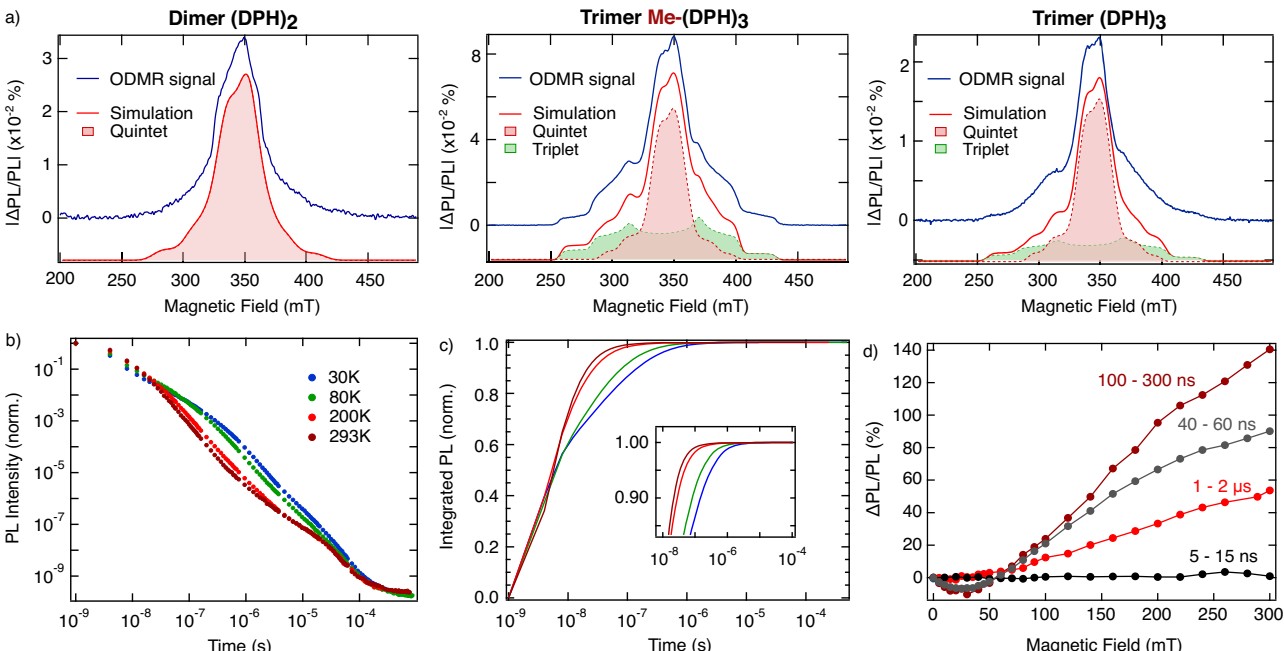

**Fig. 2 | Spin-dependent luminescence. a** ODMR spectra (blue) including simulation (red) for $(DPH)_2$, Me-$(DPH)_3$, and $(DPH)_3$. Whilst the trimers show involvement of both quintet (red filled) and triplet (green filled) in the luminescence, the dimer only reveals quintets. Performed at $T = 80$ K and $\lambda = 365$ nm. **b** Temperature dependent PL decay of Me-$(DPH)_3$, exhibiting long time delayed fluorescence.

Performed at $\lambda = 355$ nm. **c** Integrated PL from (**b**), revealing virtually all luminescence decayed by 2 μs. **d** Time-dependent magnetic-field dependent PL of Me-$(DPH)_3$. The spin-dependent luminescence starts at 40 ns, reaching a maximum at 100–300 ns. Performed at $\lambda = 375$ nm and $T = 80$ K (temperature dependence in Supplementary Fig. 9, full PL spectra in Supplementary Fig. 10).

triplet states. The pronounced HF signal (Supplementary Fig. 4) additionally confirms the involvement of the quintet state.

The possible involvement of the $^3$TT $\leftrightarrow {}^3$(T...T) states remains an open question for different singlet fission systems. Our magnetic resonance results suggest that their population and role may strongly depend on the extent of exchange coupling. In the dimer, only quintet states are observed in trEPR, with no corresponding triplet ODMR signal and no magnetic field effect. This is consistent with the absence of a $^n$(T...T) $\leftrightarrow T_1 + T_1$ population in this system, likely due to the short linker and rigid geometry preventing significant spatial separation between chromophores. Thus, the dimer appears to reside predominantly in the strongly exchange-coupled regime, where the only triplet signal detected by trEPR arises from ISC and does not contribute to luminescence. Therefore, the $^5$TT state is the only high-spin state participating in luminescence for the dimer.

In contrast, triplet features do appear in ODMR in the trimers, and their spectral shape provides insight into their origin. In particular, the Me-$(DPH)_3$ triplet spectrum reveals only a characteristic $m_s = 0$ over-population pattern in trEPR, which is a known signature of spin-polarized triplets formed from weakly-coupled $^n$(T...T) $\leftrightarrow T_1 + T_1$ states (see Fig. 1a). Given the close proximity of the chromophores, such triplets could eventually recombine via short-range encounters, establishing a dynamic equilibrium that yields emissive and non-emissive triplet configurations. Especially at 80 K, weakly-coupled triplet pairs persist long enough and remain sufficiently spin-polarized before subsequent triplet-triplet annihilation (TTA) that feeds population back into $^1$TT and, via thermal activation, into the emissive singlet state. These triplet encounters can appear in ODMR, even without full spatial separation, particularly shown for geminate $^n$(T...T) $\leftrightarrow T_1 + T_1$ TTA at low temperatures[41–43]. We note that $^3$TT states may also transiently arise in this context, as they are a product of geminate TTA and intermediate spin-relaxation processes during triplet-triplet encounters. The dimer, which does not access a persistent weakly coupled regime, shows no triplet ODMR features even at low temperatures, further supporting the assignment to geminate TTA from $^n$(T...T) $\leftrightarrow T_1 + T_1$ states.

Interestingly, ODMR at room temperature shows persisting quintet signal but vanishing triplet contribution (see detailed discussion below), indicating that the lifetime of the weakly-coupled triplet pairs until following encounter becomes too short to retain detectable spin polarization under ambient conditions. This is consistent with previous ODMR studies, where TTA-derived triplet resonances are detected only at cryogenic temperatures, due to the much shorter encounter lifetimes at higher temperatures[40,42,43]. In contrast, $^5$TT remains sufficiently long-lived at all temperatures and is therefore the only high-spin species detected by ODMR at room temperature (further explanation about ODMR signals in Supplementary Note 7).

These insights are further supported by temperature-dependent transient photoluminescence (trPL) data. The temperature-dependent trPL data of Me-$(DPH)_3$ (Fig. 2b) show a persistence of delayed PL for up to hundreds of microseconds, confirming the participation of high-spin states. The temperature dependence of the PL shows, whilst maintaining the same PL spectrum with time (Supplementary Fig. 5), a multicomponent decay with reduced dynamics upon temperature decrease. The behaviour is reminiscent of previous observations in MOF systems with a similar temperature dependence of multi-component decay in nsTA, where reduced conformational motion at low temperatures was found to slow down spin state interconversion and decay[8]. This is consistent with the ODMR observations, where low temperatures stabilize weakly-coupled triplet pairs, while at room temperature triplet-triplet encounters occur on shorter timescales, limiting their detectable steady-state lifetime. The PL spectral shape changes slightly with temperature in Me-$(DPH)_3$ (Supplementary Fig. 6), with a small reduction in vibronic structure and $S_0$-$S_2$ intensity at higher temperatures, suggesting increased conformational disorder that can subtly influence triplet-pair coupling as seen in carotenoids[44].

Integrated PL (Fig. 2c) shows that 99.99% of the emission occurs within 2 μs at all temperatures. In line with the previous discussion, higher temperatures reach this threshold faster. EPR data presented earlier revealed quintet states decaying within 2 μs, consistent with negligible PL after the same time, suggesting that quintet states

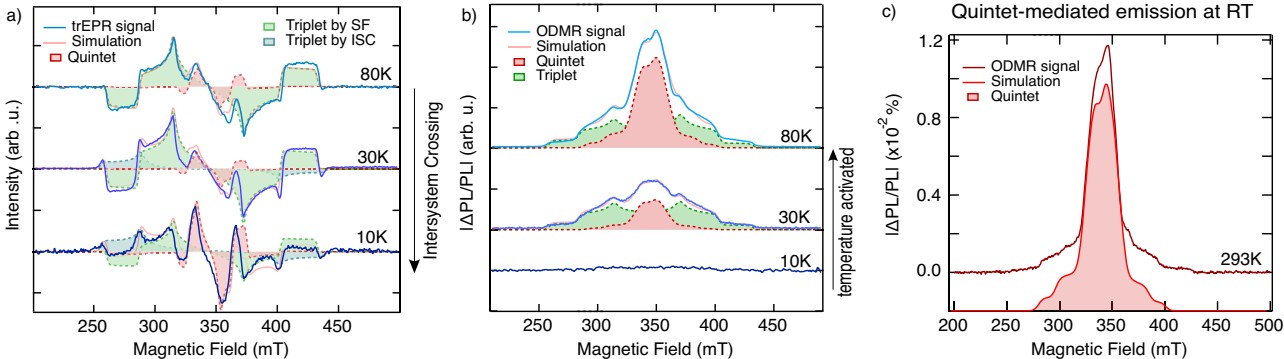

**Fig. 3 | Temperature dependence of Me-(DPH)₃. a** Temperature-dependent trEPR spectra (0–500 ns) including simulation (light red). Quintet (red filled) and SF-derived triplet (green filled) formation is present at all temperatures, whilst the ISC contribution (blue filled) increases with decreasing temperature. **b** Temperature-dependent ODMR spectra including simulation (light red). Whilst at 80 K the data show involvement of quintet (red filled) and triplet (green filled) states in the PL, the ratio decreases, and no high spin states are visible at 10 K. **c** Room-temperature ODMR spectrum (dark red), including simulation (light red), exhibits only quintet states as high-spin states.

predominantly contribute to the delayed PL at room temperature, confirmed by ODMR results at 295 K (see below). Notably, all three oligomers, including (DPH)₂ without persistent population of weakly-coupled triplet pairs, show a comparable trPL decay at room temperature (Supplementary Fig. 7). Whilst they all show equivalent generation and emission of quintet states, the similarity of trPL in all oligomers at ambient conditions hints that emissive products of triplet encounters contribute only a minor fraction to the luminescence at ambient conditions.

The nature of the intermediate triplet pair states in Me-(DPH)₃ can be further probed via PL under applied magnetic field (Fig. 2d). The MFE and the underlying mechanisms in DPH crystals and films have been extensively explored in previous studies[45–47]. The zero-crossing at 50 mT in Me-(DPH)₃ is comparable to the zero-crossing typically seen in inter-molecular DEH-DPH systems (Supplementary Fig. 8), with both exhibiting the same zero-field splitting value $D$[24]. However, whilst the overall MFE shape in Me-(DPH)₃ is similar to the previously reported MFE in DPH, variations lie in the time dependence. The initial 15 ns show no magnetic field dependence, followed by emerging effects at 30–40 ns. This agrees with a magnetic-field-independent decay of nsTA (Fig. 4b, see below), indicative of initially strongly-coupled triplet pairs formed and contributing to PL. After 40 ns, the system exhibits a MFE contrast, increasing to up to 140% for 100–300 ns after photoexcitation, proving that the weakly-coupled triplet pair regime is entered. This aligns with the EPR measurement of the trimer, which reveal an overpopulation of the $m_s = 0$ triplet sublevels, characteristic of spin-polarized triplet originating from weakly-coupled ⁿ(T...T) ↔ T₁ + T₁ states. The MFE effect results from suppressed spin-mixing by introducing Zeeman splitting, reducing the formation of T₁ states and leading to an increase in radiative decay via singlet states (further discussion MFE effect can be found in Supplementary Note 8)[48]. Notably, the MFE response diminishes significantly at 1–2 μs, at later times becoming nearly undetectable (Supplementary Fig. 9). This overlaps with the lifetime of ⁵TT states that feeds back into the emissive singlet state, and suggests a dynamic equilibrium between ⁵TT, ¹TT and ⁿ(T...T) states with ongoing spin evolution into the triplet-pair manifold during this period. The persistence of a MFE effect up to room temperature but with faster decay kinetics (Supplementary Fig. 9) indicates that the weakly-coupled regime is accessed even under ambient conditions but depopulates more rapidly.

## Temperature-dependence of quintet and triplet states

To investigate temperature-dependent quintet and triplet formation, and access energetics between these states, we conducted temperature-dependent magnetic resonance measurements for Me-(DPH)₃. Figure 3a present the temperature-dependent EPR analyses of

quintet and triplet state formation, while ODMR measurements (Fig. 3b, c) illustrate the temperature dependence of optical readout.

Temperature-dependent trEPR (Fig. 3a) evidences quintet and triplet formation for all temperatures down to 10 K, with triplet-to-quintet ratio decreasing towards 10 K, consistent with reduced diffusion and slower triplet dynamics upon temperature decrease. We showed quintet formation and triplet multiplication via SF pathway at 80 K in Me-(DPH)₃ (Fig. 1a). However, lowering the temperature induces triplets generated by ISC, resulting in spectral features that resemble those of unmethylated (DPH)₃ (see simulations in Fig. 3a and Supplementary Table 2). It has previously been well established in carotenoids that methyl groups perturb the planarity of the conjugated region[28] as well as undergo internal rotations that are still present in frozen toluene[33,49]. However, it is observed that at temperatures below 77 K, as seen in the trEPR spectrum at 30 K (Fig. 3a), the rotation of methyl groups is constrained, hindering possible ter-ahertz motions supporting ¹TT – ⁵TT mixing and increasing the proportion of ISC[33,50]. Also (DPH)₃ shows quintet and triplet formation via SF multiplication and ISC at 10 K (Supplementary Fig. 11), further confirming that multiexciton generation remains operative at cryogenic temperatures in this class of materials.

The optical back-pathway exhibits a stronger temperature-dependent activation, as shown in temperature-dependent ODMR (Fig. 3b, see Supplementary Table 4 for simulation parameters). The quintet-to-triplet ratio decreases as the temperature is reduced towards 30 K, and no high-spin states are observed at 10 K. However, trEPR confirms high-spin formation at 10 K, indicating a temperature-dependent activation to the emissive singlet state, which is reached at 30 K. The quintet state remains thermally activated up to 80 K, suggesting the need to overcome an additional energy barrier, the exchange interaction. As the quintet does not require thermal activation for formation but does need an additional temperature activation up to 80 K to fully participate in luminescence, it implies a positive exchange interaction of around 1–2 meV.

Notably, at room temperature, we identify only persistent quintet states as the emissive high-spin species in the trimer Me-(DPH)₃ (Fig. 3c). The presence of a measurable ODMR signal confirms that these ⁵TT states are sufficiently long-lived to support spin-selective luminescence and optical detection. Weakly-coupled triplet pairs that are detected at low temperatures may still be transiently accessed at room temperature, but their lifetimes become too short to retain spin polarization and contribute measurably to ODMR. As shown in Fig. 3c, these long-lived quintet states participate in the luminescence of Me-(DPH)₃ by a pathway to the emissive singlet state. Since direct ⁵TT emission would require a spin flip, and no spectral shifts in PL are

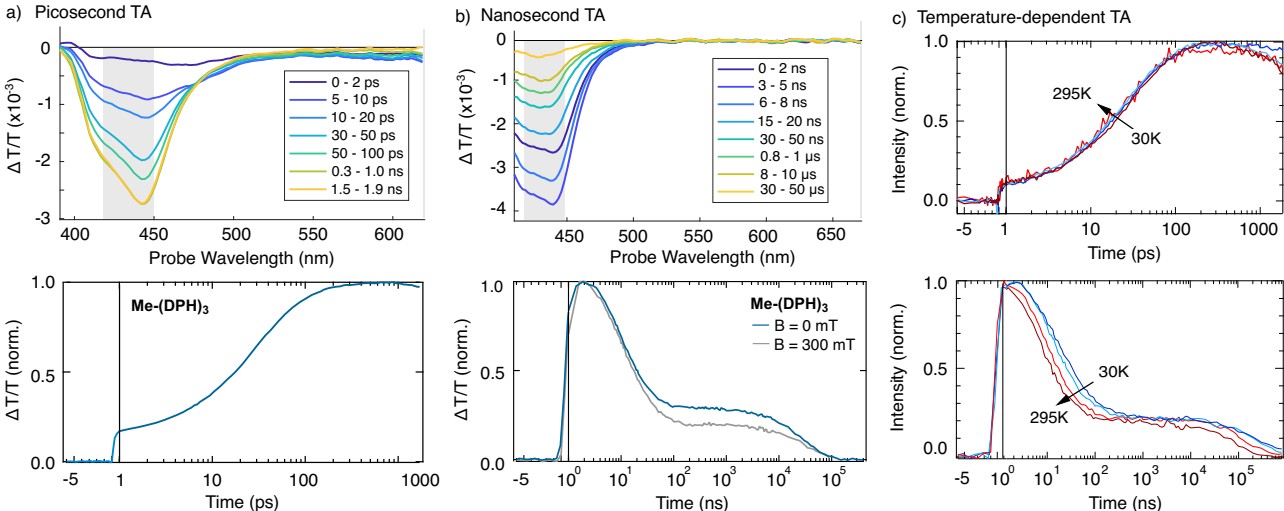

**Fig. 4 | Transient absorption spectroscopy of methylated DPH trimer Me-(DPH)₃.** **a** TA spectra (top) and kinetics (bottom) at room temperature of Me-(DPH)₃ in the picosecond regime revealing ultrafast formation of ⁿ(TT) states. **b** TA spectra (top) and kinetics (bottom) at room temperature of Me-(DPH)₃ in the nanosecond regime revealing decay of ⁿ(TT) and long-lived T₁ states. Gray shaded areas in **a** and **b** indicate the spectral regions used for extraction of the kinetics. **c** Temperature-dependent TA kinetics in picosecond (top) and nanosecond (bottom) regime. PsTA shows no change upon temperature increase, while nsTA exhibits faster decay with increasing temperature, but long-lived triplets for all temperatures.

observed over time, the reverse counterpart of the forward singlet-fission process enables a spin-allowed back pathway that feeds population back into the emissive singlet state (Supplementary Fig. 17).

This result provides direct experimental evidence of optically detected quintet-state emission in a singlet fission system at room temperature. Importantly, the long-lived nature and optical addressability of the quintet state enables optical initialization, coherent spin manipulation (Rabi oscillations, Supplementary Fig. 1), and optical readout – establishing Me-(DPH)₃ as a promising platform not only for singlet fission, but also for quantum technology applications.

### Triplet-pair dynamics in Me-(DPH)₃ via ultrafast spectroscopy

Since the initial step of singlet fission is happening in timescales of pico- to nanoseconds, we use picosecond (ps) and nanosecond (ns) TA to confirm the fast triplet pair formation and decay in the here introduced Me-(DPH)₃. Furthermore, temperature-dependent TA can timely resolve the temperature-activation to the singlet state probed by magnetic resonance measurements before.

While ultrafast spectroscopy data of (DPH)₂ and (DHP)₃ were discussed previously[25], we demonstrate the rapid formation of TT states also via psTA in Me-(DPH)₃ (Fig. 4a). Upon excitation with a 400 nm pump pulse of 200 fs duration, a broad photoinduced absorption (PIA) is immediately observed, which diminishes, concomitant with the growth of an intense higher energy PIA within a few tens of picoseconds. The initial PIA is typically associated with S* transitions, with S* denoting the singlet state in the DPH literature, based on rapid equilibration of the S₁ and S₂ states on sub-picosecond timescales[51,52]. The rapid emergence of the higher energy PIA is ascribed to the fast generation of strongly-correlated triplet pair states with singlet character, ¹TT, with a time constant of τ = 36 ps (see Supplementary Note 9 for further kinetic modelling)[53,54]. Nanosecond TA (Fig. 4b) initially reveals the decay of the TT state with a time constant of τ = 12 ns, followed by the appearance of a long-lived plateau. Based on magnetic field-dependent TA (grey), and on MFE response (previously shown in Fig. 2d), we attribute the initial, magnetic-field-independent decay to TT states undergoing either fusion to the ground state or triplet pair separation to ⁿ(T...T), followed by a magnetic field-dependent plateau. Similar conclusions were reached in our previous study of DPH oligomers based on optical studies, where the

plateau was assigned to isolated triplets formed via relaxation pathways from weakly-coupled ⁿ(T...T) states, based on spectral similarity to sensitized triplets[25]. Those criteria remain compatible with a contribution from isolated triplets, whilst the present spin-sensitive measurements also reveal a contribution from long-lived high-spin states such as ⁵TT that persists within the plateau until a few μs and can feed population into other triplet pair configurations. Thus, the long-lived component in Me-(DPH)₃ likely reflects a mixed contribution of multiexcitonic species of weakly exchange-coupled pairs, followed by triplet encounters and their products, and isolated triplet excitons – consistent with both optical and magnetic resonance studies.

Figure 4c shows temperature-dependent TA measurements to assess temperature barriers. The initial rise (top panel) is temperature-independent, consistent with an energy-barrierless formation of the ¹TT states. In contrast, the subsequent decay (bottom panel) shows a temperature dependence, indicating a thermally activated process towards room temperature. This temperature dependence matches the behaviour observed in the temperature-dependent magnetic resonance data (Fig. 3), supporting a mechanism where the forward process of ¹TT state formation is temperature-independent, as verified by Fig. 3a. However, ¹TT states undergo back-conversion to a higher-energetic S* state, requiring a temperature-activation, comparable to Fig. 3b. These observations confirm that singlet fission in this system is exothermic, implying thermally activated triplet fusion.

## Discussion

Based on the comprehensive set of spin-sensitive measurements performed, we develop a coherent picture of high-spin state dynamics in Me-(DPH)₃ and related oligomers (Fig. 5). Unlike previous reports that inferred quintet emission indirectly or required cryogenic conditions, our data provide direct observation of quintet-mediated emission at room temperature in solution-processable organic oligomers. This not only confirms the presence of optically addressable high-spin states but also opens the pathway towards applications in real-world devices and quantum photonic technologies. The excited-state formation dynamics can be understood in terms of a continuum of interconverting spin states:

The formation of spin-correlated triplet pairs can be tracked following photoexcitation, where strongly exchanged-coupled triplet

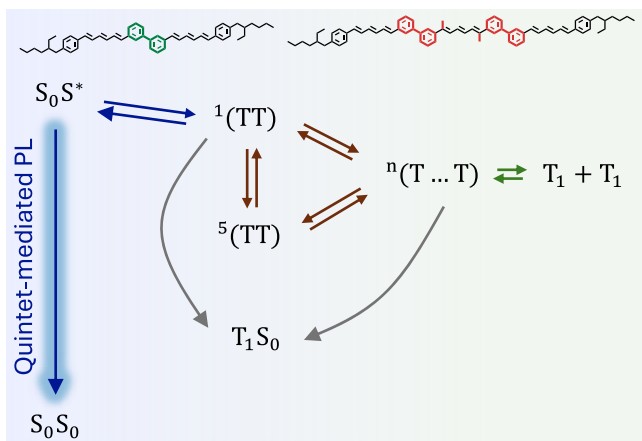

**Fig. 5 | Overview of high-spin state generation pathways and the optical back-pathway.** Photoexcitation rapidly forms $^1$TT states, which interconverts with $^5$TT states via transient exchange fluctuations. The extended system of the trimers allows weakly-coupled triplet states to be formed $^n$(T...T) $\leftrightarrow$ T$_1$ + T$_1$, which eventually recombine via TTA and form a dynamic equilibrium. Quintet states act as a central long-lived (-2 μs) species and participate in luminescence in all oligomers up to room temperature. $n = 1$, 3 or 5 for singlet, triplet or quintet configurations, respectively.

pairs $^1$TT are formed rapidly and without an activation barrier. Temperature-dependent but magnetic-field independent nsTA and PL decaying within the next 40 ns are consistent with strongly-bound $^n$TT states that primarily decay via the singlet pathway. Quintet state formation is confirmed in both the dimer and the trimer by trEPR. However, only the trimer exhibits a magnetic field effect in TA and PL beyond 40 ns, indicating that the extended system accesses a weakly exchange-coupled regime. This is further corroborated by an $m_s = 0$ overpopulation pattern in trEPR, confirming actual occupation of $^n$(T...T) states. The MFE contrast peaks and decays within the same time as the quintet population (-2 μs), implying a dynamic equilibrium between $^1$TT, $^5$TT, and transient $^n$(T...T) $\leftrightarrow$ T$_1$ + T$_1$ encounters. Thus, $^5$TT acts as a central, long-lived species that can interconvert with other multiexciton configurations, particularly in the extended trimer system. In contrast, the dimer shows quintet formation in EPR without a magnetic field effect in PL, suggesting $^1$TT $\leftrightarrow$ $^5$TT conversion via transient exchange fluctuations, rather than persistent occupation of weakly-coupled states. This is supported by the presence of only ISC-derived triplets in the trEPR spectrum of the dimer, indicating no formation of weakly-exchanged coupled triplet pairs via SF pathway.

The emissive behaviour of the systems can be resolved by combining PL and ODMR. Integrated PL shows that 75–90% emission within the first 40 ns, dominated by prompt and early delayed fluorescence from strongly coupled pairs decaying via S*. At later times, spin-dependent delayed fluorescence arises from paramagnetic states that retain access to an emissive back-pathway as reflected in the ODMR response. At 80 K, the dimer exhibits only quintet emission in ODMR, proving an optical readout channel from $^5$TT via the spin-allowed back pathway to the emissive singlet state. In the trimers, both triplet and quintet states contribute to the ODMR signal at low temperatures—consistent with the presence of $^n$(T...T) occupation. Given the chromophore proximity, such triplets can recombine via short-range encounters, establishing a dynamic equilibrium between emissive and non-emissive triplet configurations. At room temperature, however, triplet-based emission becomes negligible in ODMR: dissociated triplet pairs recombine too rapidly to maintain spin polarization or contribute significantly to PL, in agreement with TTA-related ODMR studies only performed at low temperatures[40,42,43]. Whilst the contribution of triplets to emission depends on both temperature and chromophore number,

quintets in delayed emission are observed under all conditions. This demonstrates that $^5$TT is not only formed in all oligomers but also remains optically addressable throughout the temperature range from 30 K to room temperature via a back-pathway to the emissive singlet state. At room temperature, long-lived $^5$TT states (up to -2 μs) constitute the dominant emissive reservoir, providing a robust platform for optical spin readout and enabling coherent spin operations in solution-processable SF materials (further information about quintet-mediated emission pathway is given in Supplementary Note 10).

The structural influence on high-spin pathways becomes evident in the distinct behaviour of Me-(DPH)$_3$, compared to (DPH)$_3$ and (DPH)$_2$, by exclusively showing quintets and weakly-coupled triplet pairs above 80 K. This makes it the only oligomer exhibiting the ideal pathways characteristic of a SF molecule in contrast to linear (DPH)$_3$ that demonstrates an additional triplet generation pathway via ISC. In contrast, the dimer (DPH)$_2$ does not generate persistent weakly-coupled triplet pairs, which is in agreement with previous iSF studies demonstrating that the ability to decouple $^1$TT into independent triplets is highly sensitive to the nature of the linker and the overall geometry of a dimer[55–57]. Methylation of the central DPH unit in Me-(DPH)$_3$ shows a slight net blue-shift and vibronic reweighting of the S$_0$–S$_2$ absorption, consistent with methyl-induced torsion that slightly reduces effective conjugation and electronic delocalization (Supplementary Fig. 13). These methyl groups subtly modulate the local geometry by reducing planarity and introducing low-frequency torsional motions. Such motions can enhance $^1$TT–$^5$TT mixing relative to ISC, consistent with negligible ISC contributions at 80 K[33,50].

Our findings demonstrate that increasing the number of chromophores and adjusting their spatial arrangement influences the degree of triplet–pair coupling, highlighting the critical role of molecular architecture in governing spin-state formation and achieving complete SF functionality. Despite differences in triplet formation, we demonstrated the formation and the underexplored emission via quintet states in all oligomers, highlighting this pathway as a universal feature of all presented structures. Remarkably, emission via quintet states is verified by ODMR even at room temperature, a phenomenon never reported in the literature for SF systems. Notably, the observation of quintet-state emission at room temperature—and its detection via ODMR—confirms that these states are both long-lived and optically addressable under realistic conditions. The ability to initialize, manipulate, and detect spin states entirely optically establishes Me-(DPH)$_3$ as a promising system for exploring optically addressable high-spin dynamics relevant to emerging quantum and optoelectronic materials.

In conclusion, we have shown how the molecular architecture of DPH dimers and trimers controls high-spin state formation and enables quintet-mediated emission in intramolecular singlet fission. Leveraging a unique set of spin-sensitive techniques, we established a unified picture of interconverting paramagnetic spin states, identifying the conditions under which quintet and weakly-coupled triplet pairs are accessed and participate in luminescence. We unambiguously distinguished between formation of weakly exchange-coupled triplet pairs and triplet excitons generated by ISC, identifying the methylated trimer Me-(DPH)$_3$ as the only oligomer that suppresses ISC-derived triplets above 80 K and thus provides structure-dependent control of spin-state formation. We reveal that quintet states are not only generated in all studied oligomers but also provide an optical readout pathway in the whole temperature range from 30 K—dominating delayed fluorescence even at room temperature. This unprecedented finding is not only of great importance for the SF community, but also for molecular quantum technologies where quintet states can serve as optically addressable high-spin reservoir. The ability to control high-spin state formation via molecular design and to probe quintet-mediated emission up to room temperature represents an important advance, opening new opportunities for developing emissive high-spin materials for optoelectronic and quantum applications.

## Methods

### Electron paramagnetic resonance

Transient EPR (trEPR) was performed at a Bruker E580 EleXSys spectrometer using a Bruker ER4118-MD5-W1dielectric TE01δ mode resonator (around 9.70 GHz) in an Oxford Instruments CF935 cryostat. Pulsed optical excitation was performed with an Ekspla NT230 Laser with pulse length of 3 ns, and repetition rate of 50 Hz, using pulse energies of 1.0 mJ at 355 nm. By sweeping the magnetic field, two-dimensional data sets are recorded with Bruker software Xepr, where trEPR spectra are extracted at different time intervals after laser excitation. The amplifiers for pulsed EPR (Applied Systems Engineering) had saturated powers of 1.5 kW, where Rabi notations were collected at different delays after flash (DAF) recorded by Bruker software Xepr.

### Optically detected magnetic resonance

ODMR experiments were carried out using a modified X-band spectrometer (Varian) equipped with a continuous-flow helium cryostat (Oxford Instruments CF935 cryostat) and a home-build optical resonator (based on Bruker ER4118-MD5-W1) with optical access. Optical irradiation was performed with a 365 nm LED (M365L3 Thorlabs) from one side-opening of the cavity. PL was detected with a silicon photodiode on the opposite opening, using a 375 nm long-pass filter to reject excitation light. The PL signal was amplified by a current/voltage amplifier (Femto DHPHA-200) and recorded by a lock-in detector (Stanford Research Systems SR830) referenced by on-off modulation of microwaves with a frequency of 517 Hz. The change in PL upon magnetic resonance conditions ΔPL, was directly divided by the total PL to give the ODMR contrast of ΔPL/PL.

### Transient absorption spectroscopy

Transient absorption experiments were conducted on a setup pumped by a Ti:sapphire amplifier (Solstice Ace, Spectra-Physics) emitting 200 fs pulses centred at 800 nm at a rate of 1 kHz and a total output of 7 W. The pump for sample excitation at 400 nm was provided by the second harmonic of the 800 nm output. To collect picosecond dynamics in the ultraviolet range, the 800 nm fundamental output of the amplifier was used to generate 380–620 nm probe by focusing the 800 nm fundamental beam onto a $CaF_2$ crystal (Eksma Optics, 5 mm) connected to a digital motion controller (Mercury C-863 DC Motor Controller), after passing through a mechanical delay stage. The transmitted pulses were collected with a single-line scan camera (JAI SW-4000M-PMCL) after passing through a spectrograph (Andor Shamrock SR-163).

For the nanosecond dynamics, the probe beam was generated with a supercontinuum laser (LEUKOS Disco 1 UV) and was delayed electronically with respect to the pump. The pump and probe beams were overlapped on the sample and focused into an imaging spectrometer (Andor, Shamrock SR 303i). The beams were detected using a pair of linear image sensors (Hamamatsu, G11608) driven and read out at the full laser repetition rate by a custom-built board from Stresing Entwicklungsbüro.

### Transient photoluminescence

Transient PL spectra were collected using an electrically gated intensified CCD (ICCD) camera (Andor iStar DH740 CCI-010) coupled with an image identifier tube after passing through a calibrated grating spectrometer (Andor SR303i). Samples were excited using pump pulses obtained from a home-built narrowband NOPA whilst a suitable longpass filter (375 nm or 425 nm) avoided scattered laser signals entering the camera. The kinetics were obtained by setting the gate delay steps with respect to the excitation pulse. The gate widths of the ICCD were 4 ns, 10 ns, 250 ns, 1 μs and 10 μs, with overlapping time regions used to compose decays. Temperature-dependent measurements were performed using a closed-circuit pressurized helium cryostat (Optistat Dry BL4, Oxford Instruments), a compressor (HC-

4E2, Sumitomo) and a temperature controller (Mercury iTC, Oxford Instruments).

### Time-resolved magnetic field effect

Time-resolved MFE measurements were performed with the similar electrically gated intensified CCD (Andor iStar DH334T-18U-73) camera as in trPL, whilst the excitation was performed by Q-switched Nd:YVO4 laser (Picolo AOT 1, Innolas) equipped with integrated harmonic modules for frequency-tripling (to 355 nm) provided pump pulses (500 Hz, <800 ps nominal FWHM). The gate width of the ICCD was fixed for one time-resolved MFE trace at the given gate, whilst the magnetic field was swept in 5–20 mT steps. For magnetic field and temperature control, a magnetocryostat system (Montana He cryostat, 3.4–350 K) with a Magneto-Optic module (cryostation s50 · MO) was used. Three measurements were taken at 0 mT, as this is the reference measurement used to calculate the magnetic field effect.

### Sample preparation

Solution samples were diluted in toluene (200 μM). EPR and ODMR samples were prepared by multiple freeze-pump-thaw cycles in a EPR tube (3.9 mm OD). Thin film samples were prepared by molecules dispersed in polystyrene (0.1 wt%). For EPR and ODMR, multiple films were stacked in a EPR tube and flame-sealed under vacuum. Dilute solution and dispersed films were verified to give same signals, please refer to Supplementary Fig. 12. Magnetic resonance experiments were performed either on frozen toluene solution samples (<100 K, i.e. Figs. 1, 2a and 3a, b) or polystyrene films (>150 K, i.e. Fig. 3c). This is due to instrumental constraints and the need to maintain molecular rigidity during detection of spin-polarized states. Optical measurements at room temperature were performed on solution (i.e. Fig. 4a, b) whilst cryogenic measurement were performed on polystyrene films (i.e. Figs. 2b–d and 4c), as required by the optical cryostats that operates via a cold finger to ensure optical transmission and therefore only accommodates solid-state samples.

## Data availability

The data underlying all figures in the main text and the Supplementary Information are publicly available in the University of Cambridge Apollo Repository: https://doi.org/10.17863/CAM.123400. Source data are provided with this paper.

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

## Acknowledgements

We acknowledge funding from the Physics of Sustainability and the Engineering Physical Sciences Research Council (EP/W017091/1 and EP/V055127/1) and from a UKRI Frontier Research Grant (EP/Y015584/1). T.W.B. and J.C. acknowledge EPSRC for funding the cryomagnet and Lord Porter Laser Facility (EP/T012455/1, EP/L022613/1, EP/R042802/1) and EPSRC and Calico Life Sciences LLC for studentship support. A.J.G. thanks the Leverhulme Trust for an Early Career Fellowship (ECF-2022-445). W.K.M. acknowledges EPSRC (grant to CAESR, EP /L011972/1) & John Fell Fund (0007019 & 0010710). J.B. acknowledges financial support from the German Research Foundation (DFG grant number BE 5126/6-1).

## Author contributions

J.G. performed the EPR, ODMR, PL and TA measurements. S.M. synthesized the compounds. J.G. and T.W.B. performed the MFE measurements. S.D. prepared thin film samples. W.K.M. assisted with the EPR measurements. A.J.G. helped with the setup of temperature-dependent TA. J.G. and N.C.G. designed experiments and analysed data. A.S., A.J.G., S.G. and O.M. supported the data analysis. J.B., J.C., A.R., H.B. and N.C.G. supervised group members involved in the project. J.G. and N.C.G. wrote the manuscript with input from all authors.

## Competing interests

The authors declare no competing interests.
