## [Transparent Peer Review file · Nature Communications]

High-Spin State Dynamics and Quintet-Mediated Emission in Intramolecular Singlet Fission

Corresponding Author: Professor Neil Greenham

Version 0:

Reviewer comments:

Reviewer #1

(Remarks to the Author)

Although the authors have made a considerable effort to address the reviewers' concerns, I could raise several critical issues still that need to be resolved before publication.

I raised concerns about the novelty of this work and its suitability for publication in a high-impact journal. The authors have revised the Introduction section to emphasize the significance of their study, but I find the current presentation unconvincing. I recommend that the authors better acknowledge prior research in this field. There is a substantial body of literature on oligomers, including those beyond dimers, as well as relevant studies on singlet fission that explore structure–property relationships in spin dynamics via magnetic field effects. Additionally, other significant ODMR-related works should be considered. By appropriately referencing these relevant studies, the authors could more effectively highlight the novelty of their work.

To be specific, the authors frequently use the phrase "for the first time" in the manuscript to emphasize their optical detection of 5(TT) using ODMR. However, considering the previous reports by Bayliss et al. (PRL, 2014) and Joshi et al. (JCP, 2022), is this claim justified? Recent studies by Kim et al. (JACS, 2024) and Millington et al. (JACS, 2024) also investigate structure–property relationships in spin dynamics through magnetic field effects. What structural parameters distinguish the DPH compounds from those in previous studies? The planarity of the DPH backbone appears to be an interesting parameter, but its role in spin dynamics is not sufficiently emphasized. Furthermore, the interpretation presented in this work differs significantly from that of Millington et al., despite the use of the same chemical systems and the involvement of several overlapping authors. The key difference lies in the omission of the 3(TT) in the present study's interpretation—a point I elaborate on in my second comment below. Again, there is a large body of singlet fission research on trimers, tetramers, and even larger systems, too numerous to list here.

2. Previously, I raised concerns regarding the use of the term "ISC" in this manuscript and the potential role of the 3(TT) state as an intermediate. While I understand the authors' intention to describe the 1(TT) \rightarrow T₁ transition as an intersystem crossing (ISC) process, the rationale for completely excluding the 3(TT) state from their interpretation is unconvincing. According to the Wigner–Eckart theorem, the transition between 1(TT) and 3(TT) is forbidden; thus, 3(TT) can only be generated from 5(TT). In other words, 3(TT) cannot interact or mix with 1(TT) at all. As a result, 3(TT) should not be optically detectable as a differential PL signal in ODMR. The authors may argue that 3(T···T) could be formed from 1(T···T) in a weakly interacting regime, but based on the MFE trace shown in Figure 2d, I do not believe this scenario applies to the DPH systems studied here. Microwave pumping and subsequent ODMR detection of individual triplets represent a fundamentally different situation, which only applies when two triplets encounter each other in a nongeminate fashion. I also note that Reviewer 2 raised serious concerns regarding the omission of 3(TT) in the authors' interpretation. This issue must be addressed seriously with careful consideration.

3. In my comments 3 and 4, I raised questions regarding the origin of the MFE in the DPH systems studied here. However, the authors' explanation of the low-field effect (below 50 mT) is not convincing. The authors responded that "the spin character is distributed across all nine TT states at zero field" and that "subtle field-induced rephrasing and spin-spin interactions can suppress the formation or recombination of emissive singlet states, leading to a negative fluorescence MFE at low fields." This explanation does not align with the previous study by Thompson et al., which the authors referenced in their response, nor with other well-established works by Merrifield et al. (*Pure and Applied Chemistry*, 1971) and Steiner et al. (*Chemical Reviews*, 1989). At zero field, the singlet character should be confined to only three states. In the low-field regime (below 50 mT), this character can be distributed to a maximum of six states (when D1 and D2 are identical), which explains the observed decrease in PL. At higher fields (above 50 mT), the singlet character becomes restricted to just two states, resulting in a positive MFE. As Thompson et al. have pointed out, if the singlet character were distributed across all nine states at zero field, a monotonic MFE would be observed without a distinct low-field effect (<50 mT). However, this is not the case for the DPH system studied here, as evidenced by Figure 2d. Instead, the observation in Figure 2d suggests an evolution in the distribution of the singlet character, transitioning from three to six states, and then from six to two states, as the magnetic field strength increases. This scenario excludes any mixing or interaction between $1(T\cdots T)$ and $3(T\cdots T)$, indicating that the interaction occurs solely between symmetric states, $1(TT)$ and $5(TT)$.

Reviewer #2

(Remarks to the Author)
See attached.

Reviewer #4

(Remarks to the Author)
please see the attached comments

Version 1:

Reviewer comments:

Reviewer #1

(Remarks to the Author)

The revised version of the manuscript shows substantial improvement in both clarity and scientific depth compared with the earlier submission. The authors have clearly refined the description of their experimental results and significantly strengthened the mechanistic interpretation of high-spin state dynamics in diphenylhexatriene-based oligomers. The study convincingly demonstrates the formation and emission of quintet states in intramolecular singlet fission systems and provides valuable insight into how triplet and quintet pathways can be tuned through molecular architecture. The discovery that quintet-state emission dominates delayed fluorescence up to room temperature is particularly intriguing, as it highlights an unusual and rarely explored spin-mediated radiative process in organic systems. This observation not only extends the conventional understanding of singlet fission photophysics but also suggests new opportunities for exploiting high-spin states in quantum photonics and optoelectronic applications. The manuscript's focus on distinguishing between intersystem crossing and genuine singlet-fission-derived triplet-pair formation is both timely and important. The identification of the Me-(DPH)₃ trimer as the only system that cleanly follows the desired singlet fission pathway is well supported by the data and represents a meaningful contribution to the field. The authors' use of complementary spin-sensitive optical techniques is a particular strength, though the specific methods employed could be described more explicitly to aid the broader readership of *Nature Communications*. While the abstract and manuscript are much improved, a few points would benefit from additional clarification to further strengthen the work before publication. In particular, the statement that quintet-state optical emission dominates delayed fluorescence would be more convincing if accompanied by quantitative details or clear mention of the observables used to identify and compare the relevant spin states—such as time-resolved intensity ratios, kinetic rate constants, or magnetic-field-dependent emission data. Moreover, although the distinction between weakly exchange-coupled triplet pairs and quintet emission channels is conceptually interesting, the physical origin of the quintet emission remains somewhat ambiguous. It would be valuable to indicate whether this emission results from radiative recombination of correlated triplet pairs or from a separate spin-allowed relaxation channel. A concise schematic energy diagram or kinetic model in the Supplementary Information could clarify this point and provide greater mechanistic transparency. The manuscript would also benefit from a more explicit comparison to well-established singlet fission systems such as tetracene or perylene bisimide dimers, which would help position the novelty of DPH-based oligomers in the broader SF landscape. From a presentation perspective, the text is generally clear and well written, but the abstract could place slightly stronger emphasis on the conceptual significance of identifying emissive quintet states, rather than on the enumeration of molecular structures. Overall, the present version represents a notable improvement and now meets the standards of scientific quality, novelty, and presentation expected for publication in *Nature Communications*. The work is technically sound, the conclusions are well supported by the data, and the topic will likely attract broad interest from the communities studying high-spin photophysics, singlet fission, and organic optoelectronics. With minor clarifications—mainly to improve mechanistic clarity and contextual framing—the manuscript will make a strong and timely contribution. I therefore recommend acceptance after minor revision.

Reviewer #2

(Remarks to the Author)

This reviewer is fully satisfied with the revisions. Thus, the present version is worthy of publication.

Reviewer #4

(Remarks to the Author)

please see the attached file

Version 2:

Reviewer comments:

Reviewer #1

(Remarks to the Author)

The revised manuscript presents a comprehensive and elegant study on high-spin state formation and emission processes in intramolecular singlet fission (iSF) systems based on diphenylhexatriene (DPH) dimers and trimers. The authors employ a set of spin-sensitive spectroscopic techniques to convincingly demonstrate the formation of pure quintet states and their role in quintet-mediated delayed fluorescence up to room temperature. The detailed distinction between triplet-pair formation, intersystem crossing (ISC) pathways, and the selective SF dynamics in Me-(DPH)₃ provides valuable insight into spin-state control and emissive behavior in molecular systems.

After two rounds of revision, I find that the authors have thoroughly addressed the previous reviewer comments and significantly improved both the clarity and scientific rigor of the manuscript. The presentation is now clear and coherent, the experimental evidence is convincing, and the discussion appropriately situates the findings within the context of high-spin photophysics and molecular quantum applications.

In my view, the manuscript has now reached a high standard suitable for publication. It makes an important contribution to the understanding of spin dynamics and emissive quintet states in iSF materials, and I therefore recommend it for publication in its present form.

Reviewer #4

(Remarks to the Author)

I am satisfied with the changes made by the authors in response to my questions, as well as to the questions from the other reviewer. I am happy to recommend the paper for publication and look forward to a more detailed study from the authors on the role of Me groups in tuning the high-spin state dynamics in the (DPH)₃ platform.

made.

Reviewer #1 (Remarks to the Author):

Although the authors have made a considerable effort to address the reviewers' concerns, I could raise several critical issues still that need to be resolved before publication.

I raised concerns about the novelty of this work and its suitability for publication in a high-impact journal. The authors have revised the Introduction section to emphasize the significance of their study, but I find the current presentation unconvincing. I recommend that the authors better acknowledge prior research in this field. There is a substantial body of literature on oligomers, including those beyond dimers, as well as relevant studies on singlet fission that explore structure–property relationships in spin dynamics via magnetic field effects. Additionally, other significant ODMR-related works should be considered. By appropriately referencing these relevant studies, the authors could more effectively highlight the novelty of their work.

To be specific, the authors frequently use the phrase "for the first time" in the manuscript to emphasize their optical detection of $5(TT)$ using ODMR. However, considering the previous reports by Bayliss et al. (PRL, 2014) and Joshi et al. (JCP, 2022), is this claim justified? Recent studies by Kim et al. (JACS, 2024) and Millington et al. (JACS, 2024) also investigate structure–property relationships in spin dynamics through magnetic field effects. What structural parameters distinguish the DPH compounds from those in previous studies? The planarity of the DPH backbone appears to be an interesting parameter, but its role in spin dynamics is not sufficiently emphasized. Furthermore, the interpretation presented in this work differs significantly from that of Millington et al., despite the use of the same chemical systems and the involvement of several overlapping authors. The key difference lies in the omission of the $3(TT)$ in the present study's interpretation—a point I elaborate on in my second comment below. Again, there is a large body of singlet fission research on trimers, tetramers, and even larger systems, too numerous to list here.

We agree that ODMR has previously been applied to singlet fission systems at low temperatures, notably in the studies by Bayliss et al. (PRL, 2014) and Joshi et al. (JCP, 2022). These seminal works established the observation of triplet-derived ODMR signatures at cryogenic temperatures. In our revised manuscript we explicitly cite and discuss these reports in the context of the low-temperature triplet ODMR features we also observe (see revised lines 218 - 250). However, in contrast to these earlier studies, our work demonstrates for the first time room-temperature ODMR dominated by quintet ($5TT$) states in a molecular singlet fission system. To our knowledge, this room temperature quintet ODMR has not been previously shown and is the central novelty of our contribution. We also note that Reviewers 2 and 4 explicitly recognized the novelty of detecting quintet emission at room temperature, which supports the significance of this result.

In the revised manuscript, we have more explicitly clarified the structure-property relationships and how the methylated trimer differs from our previously studied DPH systems (Millington et al., JACS 2024). On the one hand, our work extends the previous structure–property discussion by examining the formation and emission of high-spin

states in the unmethylated and methylated DPH oligomers. On the other hand, the newly introduced methylated trimer provides a reduction of ISC pathways that remain active in (DPH)₃ by introducing methyl groups. In optical data (S12), methylation of the central DPH unit in Me-(DPH)₃ shows a slight vibronic reweighting of the S₀→S₂ absorption, consistent with methyl-induced torsion that slightly reduces effective conjugation and electronic delocalization. These methyl groups facilitate access to weakly coupled triplet-pair states by subtly reducing planarity and introducing low-frequency torsional motions. Such motions can enhance ¹TT – ⁵TT mixing relative to ISC, consistent with negligible ISC contributions above 80 K. This is further supported by the temperature dependent measurements, where ISC contributions in EPR only appear below 80K where methyl motions are frozen. We have now made this structural comparison and its impact on spin dynamics more explicit in the revised manuscript (lines 118-121, lines 316-320 and discussion point (iii)).

Regarding the connection to our earlier JACS work on DPH oligomers, we have clarified how the present results extend that picture. Our previous optical-only assignment of the TA plateau was based on spectral similarity to sensitized triplets and consistent kinetics, leading us to assign it primarily to T₁+S₀ products. Those criteria remain compatible with a contribution from isolated triplets and we are able to extend the picture with high-spin states that are possible to probe with the magnetic resonance measurements. The spin-resolved data show that the additionally long-lived quintet ⁵TT populations persist into the plateau and feed other triplet-pair configurations, as weakly-coupled pairs, within their lifetime of few μs. In the revised manuscript, we explicitly discuss the previous and additional insights with focus on Me-(DPH)₃ and that the beginning of the plateau plausibly reflects a mixed multiexciton manifold comprising triplet pairs, followed by triplet encounters and their products, and isolated triplet excitons – consistent with both our optical and magnetic-resonance observations. At later times, where luminescence reaches unity and ⁵TT states are decayed, only isolated triplet states remaining, in agreement with our earlier report.

2. Previously, I raised concerns regarding the use of the term "ISC" in this manuscript and the potential role of the 3(TT) state as an intermediate. While I understand the authors' intention to describe the 1(TT) → T₁ transition as an intersystem crossing (ISC) process, the rationale for completely excluding the 3(TT) state from their interpretation is unconvincing. According to the Wigner–Eckart theorem, the transition between 1(TT) and 3(TT) is forbidden; thus, 3(TT) can only be generated from 5(TT). In other words, 3(TT) cannot interact or mix with 1(TT) at all. As a result, 3(TT) should not be optically detectable as a differential PL signal in ODMR. The authors may argue that 3(T···T) could be formed from 1(T···T) in a weakly interacting regime, but based on the MFE trace shown in Figure 2d, I do not believe this scenario applies to the DPH systems studied here. Microwave pumping and subsequent ODMR detection of individual triplets represent a fundamentally different situation, which only applies when two triplets encounter each other in a nongeminate fashion. I also note that Reviewer 2 raised serious concerns regarding the omission of 3(TT) in the authors' interpretation. This issue must be addressed seriously with careful consideration.

We agree with the reviewer's suggestions of triplet encounters as origin of the ODMR signal. As already suggested in the previous revised manuscript, we assigned the ODMR signal at low T as arising from triplet encounters. In the current manuscript, we extended our explanation of the triplet ODMR signal with reference to TTA and why it vanishes at room temperature, which is in accordance to literature:

ODMR detection can happen in nongeminate and geminate triplet encounters, as previously shown in low temperature ODMR in singlet fission by Bayliss et al (PRL, 2014), and Joshi et. al (JCP, 2022). At low temperature, weakly coupled (T...T) states can persist long enough to undergo spin-selective recombination via TTA, feeding population back into ^1TT and, via thermal activation, to the emissive S^* state. Microwave irradiation in resonant conditions induces transitions between the triplet sublevel populations, altering the fraction of encounters that yield ^1TT , and thus producing the observed triplet ODMR features. This mechanism is a well established origin of triplet signal in both geminate and non-geminate TTA, as shows in singlet-fission systems at cryogenic temperatures (references above) or as we showed previously in OPV films (Grüne et al., Adv. Funct. Mater. 2022). We also note that the dimer, which does not access a persistent weakly coupled regime, shows no triplet ODMR features even at low temperatures, further supporting the assignment to geminate TTA from (T...T) states. We emphasize that this assignment does not require invoking a direct $^1\text{TT} \rightarrow ^3\text{TT}$ pathway, which is spin-forbidden by the Wigner–Eckart theorem. Instead, triplet features arise naturally from spin-polarized triplets generated during (T...T) encounters, consistent with established low-temperature ODMR studies.

However, at higher temperatures, the triplet-pair lifetime before TTA becomes too short to maintain detectable spin polarization within the ODMR detection bandwidth. This explains the absence of triplet resonances in the room-temperature spectra and why all TTA-originated ODMR spectra are measured at cryogenic temperatures. However, $^5(\text{TT})$ remains sufficiently long-lived at all temperatures and is therefore the only high-spin species detected by ODMR at room temperature. We also note that we showed earlier that high-spin states can be ODMR-visible at room temperature in related quartet ($\text{S} = 3/2$) systems when they do not recombine rapidly via annihilation pathways such as TTA (Gorgon et al, Nature 2023). To our knowledge, our present study is the first to demonstrate optical detection of quintet ($\text{S} = 2$) states in singlet fission by ODMR at room temperature, which constitutes a key novelty of this work.

Whilst we thus don't exclude ^3TT forming as a transient species in this process, we clarify the mixing mechanisms between ^1TT and ^5TT and the absence of $^1\text{TT} - ^3\text{TT}$ mixing in our response to the next question.

3. In my comments 3 and 4, I raised questions regarding the origin of the MFE in the DPH systems studied here. However, the authors' explanation of the low-field effect (below 50 mT) is not convincing. The authors responded that "the spin character is distributed across all nine TT states at zero field" and that "subtle field-induced rephrasing and spin-spin interactions can suppress the formation or recombination of emissive singlet states, leading to a negative fluorescence MFE at low fields." This

explanation does not align with the previous study by Thompson et al., which the authors referenced in their response, nor with other well-established works by Merrifield et al. (*Pure and Applied Chemistry*, 1971) and Steiner et al. (*Chemical Reviews*, 1989). At zero field, the singlet character should be confined to only three states. In the low-field regime (below 50 mT), this character can be distributed to a maximum of six states (when D1 and D2 are identical), which explains the observed decrease in PL. At higher fields (above 50 mT), the singlet character becomes restricted to just two states, resulting in a positive MFE. As Thompson et al. have pointed out, if the singlet character were distributed across all nine states at zero field, a monotonic MFE would be observed without a distinct low-field effect (<50 mT). However, this is not the case for the DPH system studied here, as evidenced by Figure 2d. Instead, the observation in Figure 2d suggests an evolution in the distribution of the singlet character, transitioning from three to six states, and then from six to two states, as the magnetic field strength increases. This scenario excludes any mixing or interaction between $1(T\cdots T)$ and $3(T\cdots T)$, indicating that the interaction occurs solely between symmetric states, $1(TT)$ and $5(TT)$.

We thank the reviewer for pointing out this important clarification. We agree that our earlier wording was imprecise in describing the field-dependent redistribution of singlet character in the triplet–triplet manifold. Following the framework of Steiner (*Chem. Rev.* 1989), and the analysis by Thompson et al. (*Phil. Trans. R. Soc. A*, 2015), the singlet character is not distributed over all nine TT states at zero field. Instead, at zero field it is confined to three states, expands to a maximum of six states in the low-field regime (<50 mT, for identical triplets), and is then restricted to two states at higher fields as the Zeeman interaction dominates. This redistribution explains the negative-to-positive magnetic field effect observed in our DPH oligomers, and is fully consistent with the lineshape in Figure 2d. The reviewer pointed out correctly, if singlet character were indeed distributed over all nine states at zero field, only a monotonic MFE would be expected, as is seen for solution-phase tetracene in the given reference. We have corrected our explanation accordingly in the revised Supporting Information.

Our ODMR and trEPR experiments are carried out in the high-field regime (X-band, ~340 mT), which is why we more focused more on the spin states in this regime. The EPR data further support a predominant mixing within the 1TT – 5TT manifold, as suggested by the magnetic field measurements, for the following reasons: The dimer does not access a weakly coupled regime, as evidenced by trEPR and the absence of magnetic-field-dependent PL, but generates quintet states. 5TT formation is enabled by subtle structural fluctuations that transiently modulate the exchange interaction J within a strongly coupled triplet pair, allowing spin evolution from 1TT to 5TT , as recently shown in macrocyclic pentacene dimers by Ishi et al. (*J. Am. Chem. Soc.* 2024). According to the Wigner–Eckart theorem, direct $^1TT \rightarrow ^3TT$ transitions are symmetry-forbidden and would only appear through symmetry breaking. It would manifest as ISC-like polarization pattern, similar to $^5TT \rightarrow ^3TT$ conversion. However, such symmetry breaking should be equally or even more pronounced in Me-(DPH)₃, yet no ISC-like signal is observed in this system. This strongly suggests that the ISC signatures detected in the dimer originate from direct ISC into T_1 , rather than from 1TT – 3TT or 5TT – 3TT conversion, also in agreement with Ishi et al, 2024.

This assignment is reinforced by the polarization patterns: the ISC signal in the dimer is identical to that of the DPH monomer, whereas $^1\text{TT}-^3\text{TT}$ (or $^5\text{TT}-^3\text{TT}$) mixing would lead to different zero-field population patterns. In $\text{Me}-(\text{DPH})_3$, the presence of methyl groups introduces torsional motions that enhance electronic-vibrational coupling, thereby facilitating efficient $^1\text{TT}-^5\text{TT}$ mixing while suppressing ISC-like signatures. The additional ISC features visible in $(\text{DPH})_2$ and $(\text{DPH})_3$ at temperatures below 80 K, when molecular motions are frozen out, are also consistent with this picture.

Reviewer #2 (Comments for the Author):

Previous question: “1) Fig.2: From c), the authors mention that the eaaaaa-pattern triplet formation is due to the reduction in the exchange interaction to the very weakly triplet pairs with selective formation of $m_s = 0$ in the individual triplets. However, the free triplet in the selective formation in $m_s = 0$ might be a result of the triplet-TTA process with $3(\text{T}1+\text{T}1) \rightarrow 3(\text{TT}) \rightarrow \text{T}+\text{S}0$ (DOI: 10.1073/pnas.1820932116) because the separated quintet $\text{T}+\text{T}$ character ($\text{Q}0$) can undergo the $\text{Q}-\text{T}$ conversion to $3(\text{T}+\text{T})$ at $m_s = 0$. Please comment on the possibility of the participation of the TTA product in the present systems especially in the trimer samples. If no triplet-TTA product is conclusive, why this TTA does not occur? What is the bottleneck of this?”

New comments to question 1:

-> Reviewer comment: The second and the third decay components in Fig. S6 can be explained by the delayed fluorescence via $\text{T}+\text{T}$ causing singlet-TTA. Even in the dimer system, equilibrium among $\text{S}1$ and TT and $\text{T}1+\text{T}1$ will induce the triple exponential decay as reported previously (J. Phys. Chem. C 2021, 125, 33, 18287–18296). Such very weakly coupled $\text{T}1+\text{T}1$ should induce the superposition of the singlet, triplet and quintet characters in the presence of the small J -coupling as shown in the above JPC2021 paper. This will thus cause the $3\text{T}... \text{T}$ population during the equilibrium among $\text{S}1$ and TT and $\text{T}1+\text{T}1$, which means that Fig. 5 is totally odd. Thus, the occurrence of the singlet-TTA does not exclude the triplet-TTA contribution for generating the isolated triplet product showing the eaaaaa-pattern.

-> Reviewer comment: This is not the reason for the absence of the triplet TTA at all, because the triplet TTA does not contribute to the emissive $\text{S}1$ generation but to the dark individual $\text{T}1$ product.

-> Reviewer comment: The absence of the MFE does not exclude the triplet TTA scenario, when the exchange coupling between the $3(\text{TT})$ and $1(\text{TT})$ characters is much larger (or much smaller) than the applied magnetic field.

-> Reviewer comment: Given that the plateau persists hundreds of microseconds and that the no delayed emission is observed, it is very likely that $3(\text{T}1+\text{T}1) \rightarrow 3(\text{TT}) \rightarrow \text{T}+\text{S}0$ is taking place in the dimer and in the oligomer systems during the equilibrium. Again, I note that the triplet TTA is not the process to emit the delayed fluorescence but to produce the dark isolated $\text{T}1$ species. Overall, I strongly recommend that the authors rewrite the

manuscript accordingly. Nevertheless, the occurrence of the triplet TTA does not exclude the quintet-(and T+T) mediated emission in the present study, as detailed above.

-> Reviewer comment: I would agree that such flexibility would cause the TTA for the luminescence from the T1+T1 to the S1 state. However, the occurrences of the singlet-TTA and the quintet-mediated TTA does not exclude the triplet-TTA to generate the long-lived plateau for the isolated triplet product.

We answer in the following collectively to the reviewer's comments to question 1 regarding the appearance of TTA and the products of ^3TT and T_1+S_0 :

We agree that TTA can occur in our oligomers, particularly seen by ODMR at low temperatures where triplet encounters are stabilized, consistent with previous observations for other singlet fission systems under cryogenic conditions (Bayliss et al. PRL 2014, Joshi et al. JPC 2022). As already suggested in the earlier revised manuscript, weakly coupled triplet pairs in the trimers can recombine via $(\text{T}\dots\text{T})/\text{T}_1 + \text{T}_1$ encounters, which appear at low temperatures in ODMR. We agree to emphasize the subsequent spin-state evolution more, and we rewrote the current manuscript to provide a more complete picture beyond formation, explicitly noting that weakly coupled triplet pairs encounter and enter a dynamic equilibrium with other multiexciton configurations (line 218-250), also including $^n(\text{T}\dots\text{T})$ with $n=1,3,5$ in Fig. 5. At higher temperatures, triplet-triplet encounters become more frequent and weakly-coupled triplet pairs recombine too rapidly to preserve detectable spin lifetime in ODMR. As long as quintet states persist – particularly at room temperature where they remain longer-lived than $^n(\text{T}\dots\text{T})$ or $\text{T}_1 + \text{T}_1$ and are thus detectable signal in ODMR – this equilibrium can continue to exchange population between spin manifolds. We explain the origin of the triplet ODMR spectrum at low temperatures, and the absence of triplet ODMR at high temperatures further in our next response to question 4.

Regarding the TA plateau, we acknowledge that products of TTA contribute to the long-lived plateau, particularly at later times where luminescence reached unity. In accordance with our previous work, where the $\text{T}_1 + \text{S}_0$ contribution is supported by the spectral similarity to the sensitized monomer, we reinterpret the first few μs of the plateau as reflecting a mixed population of multiexciton species, at least as long as ^5TT states are living and forming an equilibrium – weakly coupled triplet pairs, followed by triplet encounters and their products $^n(\text{T}\dots\text{T})$, and isolated triplet states. After few μs , where the luminescence reaches unity and ^5TT states are gone, the plateau will have only the products of isolated triplets left. As this manuscript focuses on the magnetic-resonance perspective, we included TA primarily to confirm rapid ^1TT formation, its similar decay behaviour across oligomers, and to support the temperature trends observed via ODMR and trEPR. But to clarify the plateau, we added the explanation to the revised manuscript (line 373-382).

Thus, we have revised the manuscript by clarifying that TTA and $\text{T}\dots\text{T}/\text{T}_1 + \text{T}_1$ recombination is plausible in our spatially confined oligomer systems and contribute to the long-lived dark triplet populatio. We clarified the origin of the TTA in the ODMR signal

(line 218 – 250) and extended the explanation of the TA plateau (line 373-382) including the T_1+S_0 products. We rearranged our discussion and Fig. 5 (including $^n(T\dots T)$ states with $n=1,3,5$ and T_1+S_0 states), to acknowledge TTA and to focus more on the equilibrium of high-spin states. We hope these changes make the mechanism in our oligomers more clear without deflecting the focus too far from our main result of quintet-mediated emission at room temperature that is visible in ODMR.

Additionally, we want to note that the dimer does not exhibit the eaeaea pattern in trEPR nor a magnetic field effect in PL, indicating that the dimer does not access a persistent weakly coupled $^n(T\dots T)$ or $T_1 + T_1$ regime. Instead, it likely undergoes $^1TT \leftrightarrow ^5TT$ conversion via transient J-fluctuations, without a persistent weakly-coupled triplet pair occupation, consistent with the rigid macrocyclic pentacene dimer described by Ishii et al. (J. Am. Chem. Soc. 2024). Whilst the trimers show a triplet signal at low temperatures in ODMR, the dimer does exhibit only a 5TT signal at 80K, further supporting that the trimer triplet ODMR signal arises from products of TTA as further explained in question 4 and 6.

Previous question: “4) Regime II and III (Page 11): This is a most interesting regime in the present study but is controversial in the present version. First, please explain why the $3(T\dots T)$ is omitted in the spin state mixing in the SCTP. For understanding the triplet feature in the ODMR spectra, this $3(T\dots T)$ can be a strong candidate because of highly possible equilibrium of $1(S1S0) \rightleftharpoons 1(T\dots T) \rightleftharpoons 5(T\dots T) \rightleftharpoons 3(T\dots T)$ in the weak SCTP to contribute to sublevel transitions in the $3(T\dots T)$ for the change in the PL intensity. But this reviewer did not find any description about the sources of the triplet resonance in the ODMR spectrum.”

New comment to question 4:

-> Reviewer comment: Although it would be better that the authors add the ODMR discussion, the authors do not mention the possible origin of the triplet-resonance of the ODMR at low temperature including why the triplet resonance is absent at higher temperature and in the oligomer sample. I do not think that the triplet contribution is minor at lower temperature in the ODMR spectra.

We thank the reviewer for pointing this out and agree that the origin of the triplet ODMR resonance at low temperature needs further clarification. As already mentioned in our last manuscript, we attribute the triplet feature observed in the trimers at 80 K to SF-derived weakly coupled triplet pairs that undergo geminate TTA. We agree with the reviewer that the explanation needs more expansion as well as an explanation about the absent signal at room temperature, which we now extensively discuss in the revised manuscript (line 218 – 250) as well as discussion point (ii).

At low temperature, we attribute the triplet feature observed in the trimers at 80K to weakly coupled triplet pairs that encounter each other due to the special confinement within the molecule. As noted in question 4, at 80K, the weakly-coupled triplet pairs persist long enough and remain sufficiently spin-polarized before subsequent TTA that feeds population back into 1TT and, via thermal activation, into the emissive S^* state. Geminate as well as non-geminate TTA has been seen before in ODMR in singlet-fission

systems at cryogenic temperatures (Joshi et al. J. Chem. Phys. 2022, Bayliss et al. PRL 2014). We also note that the dimer, which does not access a persistent weakly coupled regime, shows no triplet ODMR features even at low temperatures, further supporting the assignment to geminate TTA from (T...T) states.

At higher temperatures, the triplet-pair lifetime becomes too short – due to faster spin relaxation and more rapid annihilation – to maintain detectable spin polarization within the ODMR detection bandwidth. This explains the absence of triplet resonances in the room-temperature spectra. This is why all TTA-originated ODMR spectra are measured at cryogenic temperatures, as also discussed in the cited previous SF ODMR studies. In contrast, ⁵TT remains sufficiently long-lived at all temperatures and is therefore the only high-spin species detected by ODMR at room temperature. We showed earlier in radical-based systems that high-spin states can be ODMR-visible at room temperature in related quartet (S = 3/2) systems when they do not recombine rapidly via annihilation pathways such as TTA (Gorgon et al. Nature 2023).

We apologize for the misunderstanding, our earlier review response did not intend to say that the triplet contribution is minor but that our data do not support a dominant *emissive* contribution from TTA to the photoluminescence under ambient conditions: All three oligomers show nearly identical room-temperature trPL decay profiles, including the dimer (DPH)₂, which does not form T...T and therefore cannot undergo geminate TTA. A dominant emissive contribution of TTA would be difficult to reconcile with this similarity. But we agree with the reviewer that TTA can yield non-emissive products such as ³TT and ⁵TT whilst also only a fraction of excitons enter the weakly coupled regime persistently. In summary, as the reviewer pointed out, the absence of TTA-derived emission does not exclude TTA to be happening, especially in our spatially confined chromophore systems.

Previous question: “6) From the temperature dependence of the ODMR spectra (Fig. 4c), the triplet contribution is predominant at lower temperature. But there is no explanation on the source of the triplet resonance that contributes to the delayed fluorescence. Given the negative J-coupling in the strongly coupled TT, 3(TT) state should be populated rather than the 5(TT) at lower temperature in Fig. 5. Thus, I recommend to the authors to reconsider the origin of the ODMR including the effect of the equilibrium between the strongly coupled 1(TT), 3(TT) and 5(TT) which are not drawn in Fig.5. “

New comment to question 6:

-> Reviewer comment: Again, the authors still do not mention the possible origin of the triplet-resonance of the ODMR at low temperature including why the triplet resonance is absent at higher temperature and in the oligomer sample. I am not asking why the quintet TT is contributing to the ODMR, but why and what triplet-resonance is contributing to the fluorescence. First, please clarify whether the intermolecular triplet-triplet annihilation is associated with the ODMR (as in the paper in DOI: 10.1002/adfm.202212640). Please also rationalize why the 3(TT) and 3(T...T) are not accessible to 1(TT) which may probably be accessible to the fluorescent state at lower temperature, although the above paper says that the intermolecular triplet-pair can access to the fluorescent state. It is very

important to describe what is the most likely origin of the triplet-resonance of the ODMR spectrum in the Nature Comm journal for the quality and clarity of the paper for the broad readership.

As already discussed in the previous questions, we agree with the reviewer that TTA is a well-established mechanism that can give rise to triplet resonances in ODMR.

As outlined in question 4, at low temperature, weakly coupled (T...T) states can persist long enough to undergo spin-selective recombination via TTA, feeding population back into ^1TT and to the emissive S^* state. Microwave irradiation in resonant conditions induces transitions between the triplet sublevel populations, altering the fraction of sublevel encounters that yield ^1TT , and thus producing the observed triplet ODMR features. This ODMR mechanism is well established both for intermolecular TTA in densely packed OPV films (DOI: 10.1002/adfm.202212640) and intramolecular geminate TTA in singlet-fission systems at cryogenic temperatures (DOI: 10.1002/lpor.201100026, DOI: 10.1063/5.0103662). As explained in question 4, at higher temperatures, the triplet-pair lifetime becomes too short to measure within the ODMR detection bandwidth.

We note that, although $^3(\text{TT})$ and $^3(\text{T}\cdots\text{T})$ products from TTA are generally “dark” with respect to the emissive channel probed via ODMR, spin relaxation or dissociation and subsequent encounter allow re-entry into the singlet-containing manifold, enabling TTA-mediated fluorescence. However, ^1TT and $^1(\text{T}\cdots\text{T})$ products after (T...T) encounter are assumed to be the emissive species in ODMR, and reliably probing the presence of TTA. However, our comparison to the dimer indicates that this TTA channel is a minor contribution compared to the quintet pathway in the trimers, as outlined before.

Reviewer comment: Overall, this reviewer still sees again that the present interpretations of the whole data are too preliminary, although they are trying to focus on the quintet ODMR results and responded to the comments from reviewers 2 and 3. In particular, they omit the contributions of the ^3TT and $^3\text{T}\cdots\text{T}$ states (See Fig.5) which are highly odd considering the previous reports (for example, DOI: 1038/s42004-018-0008-0 in DPH samples). Thus, the present manuscript is not worthy of the publications in the Nature Comm.

We hope we have clarified the comprehensive interpretation of our dataset by expanding the discussion to TTA in alignment with our data and explained the ODMR data at low and high temperatures sufficiently. We resolved open questions about ^3TT – in general and in regard to our earlier studies – by addressing the contributions throughout the manuscript as far as our data allow as well as including the new pathways and equilibrium in Fig. 5.

Reviewer #3 (Comments for the Author):

Reviewer #4 (Comments for the Author):

The authors have extended their recent work [1] on diphenyl hexatriene (DPH) based singlet fission (SF) chromophores to now incorporate time-resolved EPR (trEPR) and optically detected magnetic resonance (ODMR) experiments. An additional methylated DPH trimer (Me-DPH₃) is also included in this paper. In the opinion of this reviewer, the key arguments about novelty are – 1.) Quintet mediated emission reported by low-temperature trEPR and nODMR (also at 300 K), 2.) a platform to selectively achieve the ⁵(TT) high-spin state from the ¹(TT) state formed by internal conversion from S* to ¹(TT). While the former demonstration is highly interesting and novel, I find the evidence presented for the latter claim unclear. I elaborate my reasons below –

a. Highly state-selective conversion of ¹(TT)₀ to ⁵(TT)₀ with absence of ³(TT) has been theoretically predicted and experimentally demonstrated in rigid, symmetric dimers. For example, see the theory work in ref. [2] and the experimental demonstration in Fig. 2 of ref. [3]. The predictability of their model provides a rational synthetic platform. Compared to this demonstration, Fig. 1 shows a large contribution from SF generated ³(TT) photoproducts. As pointed out by reviewer 3, aspects regarding the predictability of spin dynamics and a recipe for molecular design are lacking. I therefore think that [2,3] are quite relevant references in the context of the current work.

We appreciate the reviewer's insightful comments and agree that it is important to separate the discussion of (i) initial spin state formation (Fig. 1) and (ii) subsequent photoproducts arising from TTA and involving ³TT states (Fig. 2). We now explicitly cite references [2,3] (now ref. 34 and 35 in the revised manuscript) in this context, explain the arising photoproducts of TTA in ODMR (line 218 – 250) as well as adapted our discussion including Fig. 5.

Fig. 1 shows EPR and the formation of quintet and triplet states, the latter either by ISC or by formation of weakly-coupled triplet pairs. Taking the dimer as an example, the observed triplet signal can be attributed to ISC origin, leading to a population of the zero-field triplet sublevels and giving a characteristic pattern, that is only achievable by ISC (see SI, part 4). Thus, the dimer does not access a weakly coupled regime, as evidenced by trEPR and the absence of magnetic-field-dependent PL. Importantly, the dimer still produces strongly coupled quintet states. As shown in previous studies, including ref. [2] and [3], ⁵TT formation can be promoted by structural rigidity and large exchange coupling. In our case, subtle structural fluctuations are sufficient to transiently modulate J and enable spin evolution from ¹TT to ⁵TT, consistent with observations in other covalent SF dimers e.g., ref. [3] (now ref. 35 in the revised manuscript) or also recently shown in macrocyclic pentacene dimers (Ishii et al., J. Am. Chem. Soc. 2024).

As the reviewer pointed out, direct ¹TT → ³TT transitions are symmetry-forbidden but can appear through symmetry breaking. However, this transition would manifest as an ISC-like polarization pattern. Furthermore, such symmetry breaking should be equally or even more pronounced in Me-(DPH)₃, yet no ISC-like signal is observed in this system in EPR. This strongly suggests that the ISC signatures detected in the dimer originate from direct ISC into T₁, rather than from ¹TT–³TT conversion, also in agreement with Ishii et al, 2024. Nevertheless, ³TT states could still be formed transiently and decay too rapidly to

be detected in EPR. In particular, TTA will lead to transient ^3TT states upon encounter of $(\text{T}\dots\text{T})/\text{T}_1 + \text{T}_1$ states, which are visible in ODMR (Fig. 2) and we have clarified their presence in the new manuscript and discussion (lines 218 - 250; and Fig. 5). We kindly refer to question d) for the explanation of the triplet signal in ODMR (Fig. 2) related to ^3TT and TTA.

b. Further to point a), the effect of methylation on the DPH unit has been loosely connected to planarity while changes observed in the linear absorption and timeresolved photoluminescence (TRPL) measurements have been interpreted as not being significant. Fig. S5 suggests that the net oscillator strength in the PL red shifts with temperature. Similarly, Fig. S12 shows that the net absorption oscillator strength blue shifts for the methylated trimer. Such trends in the oscillator strength are interesting and might hold vital clues about what the methylation is doing to the conformational landscape and the electronic delocalization, both key for predicting spin dynamics and providing a rational synthetic recipe. Along the same lines, PL lifetime in Fig. S12b likely shows lifetime and/or %Amplitude contribution changes that may not be insignificant. Unfortunately, no further analysis of the results in Fig. S12 is presented, which I think would really help the claim about a tunable platform for high-spin state dynamics. I can understand that modeling absorption spectra and electronic structure changes may be out of the current scope but a sound discussion about the shortcomings, as mentioned above, is necessary.

Me-(DPH)₃ shows a modest blue-shift and vibronic reweighting of the $\text{S}_0 \rightarrow \text{S}_2$ absorption, consistent with methyl-induced torsion that slightly reduces effective conjugation and electronic delocalization. This behaviour is commonly seen in carotenoids and related systems as polyene-like systems where $\text{S}_0 \rightarrow \text{S}_2$ is highly sensitive to conjugation length and vibronic coupling (Hashimoto et al., Phil. Trans. R. Soc. B 2018). Importantly, despite this spectral shift, our nanosecond TA traces are indistinguishable for both trimers (Fig. S12c) where we do not measure any changes with methylation of the initial singlet/TT decay. The small offset seen in TCSPC (Fig. S12b) is probably caused by background counts and thus within the setup uncertainty; accordingly, we rely on TA as the definitive kinetic comparison.

We note that in DPH structures S_1 and S_2 lie close enough that sub-ps internal conversion produces a common singlet manifold (often summarized as S^* in the literature), so modest $\text{S}_0 \rightarrow \text{S}_2$ reweighting does not materially alter the subsequent S_1/TT decay (see e.g. Alford et al., Chem. Phys. Lett. 1982), why the TA decays are comparable.

However, the principal methylation effect appears in the spin-mixing pathway: Me-(DPH)₃ lacks the ISC contribution observed for (DPH)₃ at $T > 80$ K, while below 80 K an ISC component also appears in Me-(DPH)₃. This trend is consistent with low-frequency (terahertz) motions enhanced by methyl groups, which promote $^1\text{TT} \leftrightarrow ^5\text{TT}$ mixing and subsequent $\text{T}\dots\text{T}$ formation (Kobori et al. J. Chem. Phys. B 2020) thereby reducing the probability of ISC. This is consistent with the appearance of an ISC signal below 80K, when methyl rotations “freeze” and the THz modulations are suppressed, which diminishes ^1TT spin-evolution and increases the relative share of ISC (Wolthuis et al., Physica B+C 1985). This picture supports our claim that the platform is tuneable

primarily via high spin-state formation, rather than via changes to the early singlet/TT lifetimes.

We note that the normalized PL spectra (as shown in Fig. S5) exhibit minor vibronic changes with temperature, with the 0-1/0-2 peaks becoming slightly more pronounced at lower temperatures, but no significant spectral shift or new transitions are observed. This indicates minimal changes in conjugation length or structural rearrangement across 10-300 K, consistent with a rigid conjugated backbone. These observations align with results from carotenoid studies (Zheng et al., J. Mol. Liq. 2022), where temperature affects electron-phonon coupling and vibronic progression, but does not substantially alter absorption maxima unless large structural changes or conformational freedoms exist.

We added these differences, especially the influence of methylation on the high-spin state pathways, to various parts of the manuscript (line 118-121, 316-320) and extended the explanation about the optical data in the SI. We also included an extra discussion paragraph “(iii) Implications for SF design” in the discussion of the revised manuscript to account for the explained impact of methylation on spin state dynamics.

c. I find the discussion regarding the complete re-interpretation of the ‘triplet plateau’, presented originally in [1], to be rather vague, as was also pointed out by reviewer 3. As I understand, previous work of the authors [1] reports a hypsochromic shift in the spectral feature for independent triplets compared to 1(TT), similarity with the triplet spectrum obtained from sensitization, and similarity of ‘triplet plateau’ lifetime with that of isolated triplets in order to justify the assignment of the plateau to isolated triplets arising from SF. In the current paper, the triplet plateau is instead assigned to come from the dynamics of the quintet. I think a proper discussion about the significant re-interpretation of the results in [1] will be quite useful. It also raises a fundamental question about the (in)validity of the above criteria to identify isolated triplet dynamics based on spectral similarities and lifetimes, which is rampant in the SF community.

We agree that our earlier assignment of the TA “plateau” was based on carefully evaluated optical criteria (spectral similarity to sensitized T_1 and comparable lifetimes), and we do not aim to rebut those observations. Rather, the new spin-sensitive measurements extend that picture.

Ultrafast optics and magnetic resonance are complementary: the former provides the time resolution to follow rapid ^1TT formation and decay, while the latter supplies spin selectivity. We include TA in this manuscript primarily to confirm the fast, barrierless generation of ^1TT and its decay in Me-(DPH)₃ (on the same ps/ns timescales as in our earlier study as discussed before), and to support the temperature trends extracted from temperature dependent ODMR and EPR.

As the reviewer notes, prior optical-only work on DPH oligomers assigned the plateau to isolated triplets formed from weakly coupled (T...T) states by spectral similarity to sensitized T_1 . That procedure is common in ultrafast spectroscopy and the criteria

remain compatible with a contribution from isolated triplets in these systems. The spin-resolved data now show that long-lived quintet (^5TT) populations persist into the plateau (up to a few μs) and feed other triplet-pair configurations. Thus, the plateau need not be exclusively T_1 : it also include quintet-derived populations and their interconversion with weakly exchange-coupled $T\dots T/T_1+T_1$ configurations. Consistent with reviewer 2's points and prior low-temperature studies, geminate T_1+T_1 encounters will also populate T_1+S_0 products within this plateau.

Accordingly, the revised text states explicitly that the plateau in $\text{Me}-(\text{DPH})_3$ most plausibly reflects a mixed multiexciton manifold comprising triplet pairs, partially dissociated triplet encounters, and isolated triplet excitons – consistent with both optical and magnetic-resonance observations. At later times where the luminescence reached unity and ^5TT states decayed, the plateau can only have isolated triplets left.

Finally, our main conclusion is unchanged: quintet-mediated pathways are present in all oligomers and dominate the radiative channel at room temperature, whereas triplet-encounter signals appear in ODMR at low temperature, when their lifetimes are long enough to retain spin polarization.

d. Triplet contributions are seen at 80K suggesting significant spin relaxation within the $5(\text{TT})$ manifold and population flow into the $3(\text{T}\dots\text{T})$ manifold already at 80K (Fig. 2). It is not clear to me what is the mechanism for selective depopulation of the $3(\text{T}\dots\text{T})$ manifold into the $5(\text{T}\dots\text{T})$ at room temperature resulting in ODMR showing no contributions from the triplet at all? (Fig. 3c). The rate schematic in Fig. 5 does not consider this at all but I think this aspect is quite central to the results presented here.

We thank the reviewer for a raising that this point needs more clarity. As noted earlier, triplet-triplet encounters or spin relaxation within the $^n(\text{T}\dots\text{T})$ manifold can populate triplet states. Especially for TTA, geminate or non-geminate, it is common to detect an ODMR signal in form of a triplet spectrum. However, ODMR signal can only be detected if the population is long-lived enough, which is the case at 80 K, where triplet encounters are stabilized and the $^n(\text{T}\dots\text{T})$ state persists within the ODMR detection window.

At low temperature, we attribute the triplet feature observed in the trimers in ODMR to weakly coupled triplet pairs that encounter each other due to the special confinement within the molecule. The weakly-coupled triplet pairs persist long enough and remain sufficiently spin-polarized before subsequent TTA that feeds population back into ^1TT and, via thermal activation, into the emissive S^* state. Geminate as well as non-geminate TTA is seen before in ODMR in singlet-fission systems at cryogenic temperatures (e.g., Joshi et al. J. Chem. Phys. 2022, Bayliss et al. PRL 2014) as well as in OPV systems (Grüne et al. Adv. Funct. Mater. 2022, Shinar et al. Laser & Photonics Reviews 2012). We also note that the dimer, which does not access a persistent weakly coupled regime, shows no triplet ODMR features even at low temperatures, further supporting the assignment to geminate TTA from $(\text{T}\dots\text{T})$ states.

In contrast, at room temperature, triplet encounters happen on shorter timescales leading to T_1+T_1 and $(T...T)$ configurations recombine much more rapidly. Their residence time becomes too short to preserve measurable spin polarization within the ODMR detection window (detection window of ODMR is based on several factors, amongst others the Q factor of the microwave cavity and the use of a lock-in detector), thereby suppressing the ODMR visibility of the triplet channel at 295 K. This is why all TTA-originated ODMR spectra are measured at cryogenic temperatures, as also discussed in previous SF ODMR studies, because the triplet–triplet encounter lifetime becomes too short at higher temperatures.

In contrast, $^5(TT)$ remains sufficiently long-lived at all temperatures and is therefore the only high-spin species detected by ODMR at room temperature. As seen in trPL and magPL, its lifetime shortens with increasing temperature but stays long enough to remain ODMR-visible. We showed earlier in radical-based systems that high-spin states can be ODMR-visible at room temperature in related quartet ($S = 3/2$) systems when they do not recombine rapidly via annihilation pathways such as TTA (Gorgon et al. Nature 2023).

We integrated this important explanation in the revised manuscript (line 218-250) as well as updated Fig. 5 with forward and backward pathways.

e. To support the claim that quintet/triplet ratio decreases with temperature, it will be useful to breakdown Figs. 3a and 3b into individual quintet, triplet contributions such as that in Fig. 2a. The caption of Fig. 3 says so but I do not see that in the figure. Inferring it directly from the figure is a stretch for untrained eyes.

We have revised Fig. 3a and 3b to explicitly decompose the spectra into quintet and triplet contributions, analogous to Fig. 2a. We show now the contribution of the EasySpin simulations of the different triplet and quintet states behind the spectra (with simulation parameters in Table S2 and S4) which makes the temperature-dependent change in the quintet/triplet ratio immediately visible.

In summary, I think the argument about Me-(DPH)₃ providing a tunable platform has several shortcomings that need to be addressed. The demonstration of quintet luminescence at 300K is interesting, although the mechanism for only quintet signal in ODMR at 300K is unclear.

We thank the reviewer for the comments and acknowledging the novelty of our quintet room temperature ODMR. We trust that the revised manuscript clarifies the mechanism of triplet ODMR at low temperatures as well as the absence of an ODMR triplet signal at room temperature, given the temperature-dependent ODMR signal of TTA. The added interpretation of optical as well as magnetic resonance differences in Me-(DPH)₃ will make the positive changes upon methylation more clear to the reader. We also now frame “tunability” concretely as methylation-driven control of low-frequency torsions, which suppresses ISC above 80 K without perturbing the early S_1/TT kinetics, and maintaining the room temperature quintet formation and emission.

Reference DOIs:

1. <https://doi.org/10.1021/jacs.4c10483>
2. <https://doi.org/10.1038/s41598-020-75459-x>
3. <https://doi.org/10.1038/s41467-023-36529-6>

-

Reviewer #1 (Remarks to the Author):

The revised version of the manuscript shows substantial improvement in both clarity and scientific depth compared with the earlier submission. The authors have clearly refined the description of their experimental results and significantly strengthened the mechanistic interpretation of high-spin state dynamics in diphenylhexatriene-based oligomers. The study convincingly demonstrates the formation and emission of quintet states in intramolecular singlet fission systems and provides valuable insight into how triplet and quintet pathways can be tuned through molecular architecture. The discovery that quintet-state emission dominates delayed fluorescence up to room temperature is particularly intriguing, as it highlights an unusual and rarely explored spin-mediated radiative process in organic systems. This observation not only extends the conventional understanding of singlet fission photophysics but also suggests new opportunities for exploiting high-spin states in quantum photonics and optoelectronic applications. The manuscript's focus on distinguishing between intersystem crossing and genuine singlet-fission-derived triplet-pair formation is both timely and important. The identification of the Me-(DPH)₃ trimer as the only system that cleanly follows the desired singlet fission pathway is well supported by the data and represents a meaningful contribution to the field.

We thank the reviewer for the positive and encouraging assessment of our revised manuscript. We greatly appreciate the recognition of the improved clarity and mechanistic interpretation, as well as the significance of identifying emissive quintet states in intramolecular singlet fission systems.

The authors' use of complementary spin-sensitive optical techniques is a particular strength, though the specific methods employed could be described more explicitly to aid the broader readership of Nature Communications. While the abstract and manuscript are much improved, a few points would benefit from additional clarification to further strengthen the work before publication.

We are happy to strengthen the description and complementarity of the different techniques used. In particular, the combination of the techniques of TA, trEPR, trPL (with magnetic field) and ODMR is a strength to track the triplet and quintet formation, evolution and emission across timescales, as we describe in the following and added to the Supporting Information, section 3.

Complementarity of TA, trEPR, trPL and ODMR techniques

To measure the full sequence of spin-state formation and emission in singlet fission, we require multiple complementary techniques (Table R1). Transient absorption (TA) provides the fastest temporal resolution, directly revealing how and when triplet pairs formed which is important to prove fast triplet pair formation. However, the lack of spin selectivity requires paramagnetic techniques such as electron paramagnetic resonance (EPR). Transient EPR (trEPR) adds spin selectivity, identifying whether triplet and quintet states are occupied. An important advantage of trEPR is the ability to track the

populations mechanism of these states via singlet fission or ISC, however, the technique is usually limited to cryogenic temperatures.

Optical detection methods bridge these regimes: transient PL traces delayed fluorescence in high sensitivity up to several hundreds of microseconds. It provides the temporal decay of emissive states as well as the corresponding spectra. However, to identify which paramagnetic species are in the delayed fluorescence, optically-detected magnetic resonance (ODMR) with its spin-selectivity is required. ODMR uniquely identifies paramagnetic species (quintet or triplet) that contribute to luminescence and can be even measured up to room temperature given its good sensitivity.

By combining these complementary approaches, we connect ultrafast population dynamics (TA) with spin-resolved formation pathways (trEPR) and emissive high-spin channels (ODMR), allowing a complete description of the formation and emission of high-spin states in these singlet fission systems. The following table provides an overview about the key information provided by the individual techniques as well as the advantages and limitations.

Table R1. Overview of the complementary techniques used to probe spin state formation and emission in these singlet fission systems.

Technique	Key information provided	Advantages / Limitations
Transient Absorption (TA)	Tracks formation and decay of all photoexcited states with ps to ns resolution. Identifies fast triplet-pair formation that is unique to singlet fission.	Advantages: · Fastest time resolution (ps) · Probes both emissive and non-emissive states · Applicable at room temperature Limitations: · No intrinsic spin selectivity · Spectral overlap can limit species assignment
Transient EPR (trEPR)	Sensitive to spin-polarized paramagnetic states (quintets and triplets). Reveals population mechanism (SF vs ISC) via polarization patterns.	Advantages: · Spin-selective: identifies triplet and quintet species · Mechanism-specific information (ISC vs. SF) · Temporal information from 100 ns - μ s Limitations: · Time resolution only from 100ns onwards · Typically cryogenic · Lower sensitivity than optical methods
Transient Photoluminescence (PL)	Measures radiative decay and delayed fluorescence (ns - μ s) with very high sensitivity (performed with ICCD).	Advantages: · Directly probes emissive states · Sensitive to delayed emission kinetics and spectra. Limitations: · No spin-selectivity (partly with applied magnetic field) · Cannot distinguish triplet vs. quintet contribution without ODMR

Optically Detected Magnetic Resonance (ODMR)	Monitors PL changes under microwave resonance, identifies directly paramagnetic species (triplet or quintet) that are coupled to luminescence.	Advantages: · Very sensitive (lock-in detection) · Spin-selective for emissive paramagnetic states · Usable under continuous excitation and at room temperature Limitations: · Steady-state technique, i.e. no intrinsic time resolution. · Population mechanism inferred indirectly
---	--	---

In particular, the statement that quintet-state optical emission dominates delayed fluorescence would be more convincing if accompanied by quantitative details or clear mention of the observables used to identify and compare the relevant spin states—such as time-resolved intensity ratios, kinetic rate constants, or magnetic-field-dependent emission data.

In the revised manuscript, we now quantify the delayed fluorescence contribution associated with quintet emission by comparing time-resolved and magnetic-field-dependent PL data. The magnetic field response appears within ~40 ns after excitation, indicating that the singlet population has already decayed and the delayed PL originates from high-spin states. Integration of the PL signal beyond 40 ns shows that this delayed component accounts for approximately 10% of the total emission until 2 μ s. This timescale is in agreement with the decay of the ⁵TT population observed in trEPR and leads to the quintet ODMR at room temperature. The trPL is comparable for all three compounds at room temperature, including the dimer that does not enter the weakly-coupled regime and only has emission via quintet states. Thus, these quantitative comparisons substantiate our conclusion that quintet-state emission dominates the delayed fluorescence channel in about 10% of the total emission. We added these quantitative arguments with relevant kinetic rates to the Supporting Information, section 10, and referred to it in the main discussion.

Moreover, although the distinction between weakly exchange-coupled triplet pairs and quintet emission channels is conceptually interesting, the physical origin of the quintet emission remains somewhat ambiguous. It would be valuable to indicate whether this emission results from radiative recombination of correlated triplet pairs or from a separate spin-allowed relaxation channel. A concise schematic energy diagram or kinetic model in the Supplementary Information could clarify this point and provide greater mechanistic transparency.

We thank the reviewer for raising this important point. We now clarify in the revised manuscript that the observed emission originates from spin-allowed recombination of the ⁵TT manifold back to the singlet state, rather than from direct radiative decay of ⁵TT itself. This pathway is the spin-reverse counterpart of the forward singlet-fission process ($S_1 \rightarrow {}^1\text{TT} \rightarrow {}^5\text{TT}$), enabled by exchange-coupling fluctuations $J(t)$ that mix singlet and

quintet character and enabling the same pathway in the back direction (${}^5\text{TT} \rightarrow {}^1\text{TT} \rightarrow \text{S}_1$). Such ODMR signals from “back pathways” via the emissive singlet state are well established in related systems: for instance, triplet-mediated delayed fluorescence in donor-acceptor OLEDs via a similar $\text{T}_1 \rightarrow \text{S}_1$ conversion (Bunzmann et al., Mater. Horiz. 2020), and quartet-mediated emission $\text{Q}_1 \rightarrow \text{S}_1$ has been reported in radical-anthracene-linked systems (Gorgon et al., Nature 2023).

Importantly, we do not observe any measurable spectral shift in the delayed photoluminescence relative to prompt emission, consistent with recombination via the singlet state rather than from a distinct direct, also spin-forbidden, emissive ${}^5\text{TT}$ level. We therefore describe this process as “quintet-mediated emission”, since the quintet state acts as a long-lived reservoir that feeds the emissive singlet channel. This interpretation reconciles the time-resolved PL, trEPR, and ODMR data and we have clarified this also in the main text, line 344-347, and 449-451. In addition, we included a new complete schematic diagram including this pathway and other obtained important kinetics in the Supporting Information Fig. S17 and Section 10.

The manuscript would also benefit from a more explicit comparison to well-established singlet fission systems such as tetracene or perylene bisimide dimers, which would help position the novelty of DPH-based oligomers in the broader SF landscape.

Work on tetracene oligomers (Wang et al., Nat. Chem. 2021) has shown that increasing the number of chromophore units enables access to spatially separated, weakly coupled triplet pairs ${}^1(\text{T}..\text{T})$, which enhances free-triplet generation compared with dimers that primarily populate adjacent ${}^1\text{TT}$ states. This established a structural modulation, based on chromophore number and geometry, for the degree of inter-triplet coupling and triplet separation. Our DPH oligomers are consistent with this picture where trimers can access a weakly coupled ($\text{T}..\text{T}$) manifold at low temperature, as seen in ODMR and trEPR, whereas the dimer does not enter it persistently. However, our study goes beyond triplet-yield optimization by directly resolving and exploiting the quintet ${}^5\text{TT}$ manifold: we observe ${}^5\text{TT}$ formation in all oligomers and quintet-mediated delayed fluorescence up to room temperature, detected by ODMR.

Similar studies on perylene dimers (Hong et al, Nat. Comm. 2022) highlight that low-frequency conformational motions and interchromophore coupling influences access to distinct multiexciton manifolds. Consistent with that principle, our DPH oligomers show that methylation modulates low-frequency torsions, biasing triplet-pair mixing relative to ISC. However, our scope and outcome differ: prior perylene/tetracene studies optimized free-triplet yields but did not directly address quintet formation or emission. Here, we identify the ${}^5\text{TT}$ manifold spectroscopically and show that it feeds delayed fluorescence via spin-allowed back-transfer the emissive singlet state. To our knowledge, room-temperature optical detection of emissive quintets acting as an emissive reservoir has not been reported for tetracene or perylene oligomers.

But also in terms of triplet formation for singlet fission applications, DPH offers an important advantage: the DPH oligomers have a triplet energy of ~ 1.5 eV, well above the silicon bandgap and higher than that of tetracene (~ 1.25 eV) or perylene (~ 1.1 eV), making

it a promising singlet-fission material for silicon solar cells. Thus, DPH oligomers bridge the gap between energy-conversion systems and quantum-applications, establishing a new regime where high-spin states are not only intermediates but functional emissive states that can be used towards molecular quantum-optical applications. To position our DPH oligomers in the broader landscape, we added this comparison to the Supporting Information, Section 11.

From a presentation perspective, the text is generally clear and well written, but the abstract could place slightly stronger emphasis on the conceptual significance of identifying emissive quintet states, rather than on the enumeration of molecular structures.

We thank the reviewer for this helpful suggestion. We have revised the abstract to place stronger emphasis on the significance of quintet-mediated emission, highlighting their role as a new functional channel in singlet fission and their relevance for high-spin photophysics. The revised abstract now emphasises more the emissive quintet states up to room temperature and discuss the broader implications for spin-photon interfaces and molecular quantum applications.

Overall, the present version represents a notable improvement and now meets the standards of scientific quality, novelty, and presentation expected for publication in Nature Communications. The work is technically sound, the conclusions are well supported by the data, and the topic will likely attract broad interest from the communities studying high-spin photophysics, singlet fission, and organic optoelectronics. With minor clarifications—mainly to improve mechanistic clarity and contextual framing—the manuscript will make a strong and timely contribution. I therefore recommend acceptance after minor revision.

We thank the reviewer for this very positive evaluation and for recommending acceptance after minor revision. We have carefully implemented all requested clarifications to further strengthen the mechanistic discussion and clarity of the manuscript.

Reviewer #2 (Remarks to the Author):

This reviewer is fully satisfied with the revisions. Thus, the present version is worthy of publication.

We thank the reviewer for the positive assessment and greatly appreciate their thoughtful engagement throughout the review process.

Reviewer 4:

I have carefully revisited my previous comments from other reviewers, and the corresponding changes made in the manuscript. I unfortunately do not find sufficient evidence for the two central claims about 1. “illustrating a framework for molecular design to selectively control spin-state formation” and 2. “a design principle for DPH-based materials as platforms”.

We appreciate the reviewer's re-evaluation and have further clarified these points. The revised manuscript focuses on the experimental identification of emissive quintet ^5TT states at room temperature, a result we now emphasize explicitly in the abstract. The suppression of ISC-related pathways in the methylated trimer $\text{Me}-(\text{DPH})_3$ represents an evidence-based outcome of methyl substitution, which enables the singlet-fission process to proceed with fewer overlapping spin pathways compared to the unmethylated analogues. As both $(\text{DPH})_2$ and $(\text{DPH})_3$ exhibit ISC contributions whilst $\text{Me}-(\text{DPH})_3$ does not, the methylation influences the $^1\text{TT}-^5\text{TT}$ interaction to reduce the likelihood for ISC from ^1TT , as discussed more in the following answers.

The term "design principle" in the manuscript was intended to convey that increasing the number of chromophores and introducing methyl groups lead to measurable changes in spin-state behaviour, specifically, the absence of weakly coupled T...T states in dimers and the suppression of ISC in $\text{Me}-(\text{DPH})_3$. While we recognize that the previous wording may have implied a broader or more predictive "design framework," we have refined the text to highlight the mechanistic insights established by our experimental data in these studied systems rather than general design rules. We believe that these evidence-based findings, i.e. the demonstration of quintet-mediated emission and structure-related changes in spin-pathways of ISC, accurately reflect the scope and strength of the present study.

In response to reviewer comments, for example, comments 2 and 3 from reviewer 1, the authors now argue that both 3TT as well as intersystem crossing (ISC) born triplets may be present and contribute to ODMR signatures. The argument is then made that the only reason that room temperature ODMR does not show triplet signatures is because 5TT is long lived and the 3TT and ISC born triplet states are not. What is the mechanism behind this and therefore, whether this is a tunable property or not, is not clear from the evidence presented.

ODMR detection requires a persistent spin polarization within the experimental bandwidth, thus, only spin states with sufficiently long relaxation times (both for spin polarization and state population) contribute to the measurable signal. At low temperatures, weakly coupled triplet pairs (T...T) persist long enough before undergoing triplet-triplet annihilation (TTA), giving rise to characteristic triplet ODMR features. This is consistent with previous low-temperature ODMR studies in singlet-fission materials (Bayliss et al., PRL 2014; Joshi et al., JCP 2022). With increasing temperature, annihilation rates accelerate, and the triplet-pair lifetime before TTA becomes too short to maintain detectable spin polarization within the ODMR detection bandwidth, also in agreement with these reports. In intramolecular SF, the shorter spatial separation of the T...T pair further accelerates TTA encounters, reinforcing this temperature dependence.

In contrast, the ^5TT state is longer-lived and thus remains detectable up to room temperature. This behaviour is best described by population kinetics within the correlated triplet-pair manifold: ^1TT , ^5TT , and T...T states are in dynamic equilibrium, with ^5TT acting as a population reservoir for the delayed fluorescence. As shown by the temperature-dependent ODMR, the quintet signal increases up to ~ 80 K, exhibiting an activation energy that reflects the need to overcome an energy barrier (exchange

interaction) for conversion back to the emissive singlet state. This activation slows the decay toward luminescence and results in the long-lived ^5TT population (up to $\sim 2 \mu\text{s}$) observed in both trEPR and temperature dependent trPL. The schematic representation of the ^5TT -mediated emission pathway has been added to the Supporting Information Fig. S17.

ISC-born triplets are not observed in ODMR at any temperature, as intersystem crossing involves single-exciton conversion to a lower-lying triplet exciton that has no back pathway (through bi-exciton TTA) to the emissive singlet state. Consistent with this, the dimer, where only ISC-born triplets are formed, exhibits solely quintet ODMR, even at low temperatures. This behaviour is tuneable through molecular design in terms of chromophore number: the dimer remains in a strongly exchange-coupled regime, preventing access to weakly coupled T...T states, while increasing the chromophore number promotes occupation of weakly coupled triplet-pair states that can undergo TTA encounters. Consequently, the relative detectability of triplet and quintet features at low temperatures can be modulated by increasing the number of chromophores. Overall, the triplet ODMR behaviour is consistent with temperature-dependent ODMR studies in other singlet-fission systems whilst we highlight that the persistent room-temperature ODMR signal originates from kinetically stabilized ^5TT states.

In another example, considering the response to my comment b, the authors cite (Kobori et al. *J. Phys. Chem. B* 2020, <https://pubs.acs.org/doi/10.1021/acs.jpccb.0c07984>) for connecting the effect of methylation to the appearance of ISC triplets below 80 K but not at 300K. The authors argue that the low-frequency THz motions, are enhanced by the methyl group and promote 1TT to 5TT mixing, thus competing against ISC at room temperature. It is proposed that at 80 K these motions “freeze out” to reduce the probability of 1TT to 5TT conversion and therefore the ISC signatures become more prominent at 80 K. This argument, on which the central claim about the tunability and predictability of the DPH platform rests, seems fairly speculative to me. 80 K is $\sim 55 \text{ cm}^{-1}$. The low-frequency THz motions as discussed in Kobori et al. are only $\sim 72 \text{ cm}^{-1}$ and therefore expected to play an active role even at 80 K. This is indeed reported by Figure 5 of Kobori et al. where these motions actively participate in 1TT to 5TT conversion which is evident at 77K. Thus, “freezing out” methyl motions does not seem like a tunable knob at all.

We apologize for the misunderstanding, the low-frequency torsional modes associated with methyl groups are active at 80K, as reported by Kobori et al (*J. Phys. Chem. B*, 2020). Our experiments show that there is no ISC-related polarization in trEPR at 80K in the methylated trimer $\text{Me}-(\text{DPH})_3$, only in the unmethylated trimer $(\text{DPH})_3$. At lower temperatures, i.e. at 30K and 10K, the $\text{Me}-(\text{DPH})_3$ trEPR shows a contribution of ISC and the pattern becomes similar to that of $(\text{DPH})_3$. This indicates that at very low temperatures (30K and below), methyl motions are substantially constrained and the population of ISC-like triplets emerges also in $\text{Me}-(\text{DPH})_3$.

We clarified this temperature dependence further in the revised manuscript. Our intention was not to suggest that methyl-group dynamics constitute a directly tunable

parameter, but rather to show that this structural modification can slightly alter the relative weighting between ^1TT - ^5TT mixing to reduce ISC pathways from ^1TT . This interpretation is consistent with Kobori et al., where the THz-range modes are active at 77 K and promote ^1TT - ^5TT mixing, making our experimental findings in agreement with the vibrational mechanism proposed in that work.

Furthermore, ^5TT states dominate room temperature emission in all systems including (DPH)₃ where there is no methyl group. Thus, even the promoting role of the methyl group in ^1TT to ^5TT conversion is not clear. The above points also echo with the comments from reviewer 3 of the previous round, specifically points 2 and 3 – the unpredictability of spin dynamics and the study being focused on a methylated version of (DPH)₃ studied earlier, respectively.

All oligomers exhibit delayed emission at room temperature mediated by the ^5TT manifold. This demonstration of emissive quintet states can dominate the high-spin photophysics of intramolecular singlet fission systems up to room temperature is indeed one of the key outcomes of our study.

The methyl substitution in Me-(DPH)₃ does not generate a new emissive pathway but provides a clearer system in which competing ISC-related signatures are suppressed. The methyl groups slightly modify the exchange coupling dynamics, leading to a reduced ISC contribution and a correspondingly cleaner ^1TT – ^5TT equilibrium. All systems ultimately relax through the same ^5TT -mediated emissive channel, but Me-(DPH)₃ allows this process to be observed with fewer overlapping spin pathways in comparison to (DPH)₃. The focus on Me-(DPH)₃ is therefore intentional: its suppression of ISC pathways above 80K provides a cleaner model system in which triplet and quintet ODMR signatures can be reliably distinguished, allowing us to unambiguously identify the quintet-mediated emission that forms the central finding of this study.

Grüne et al. investigate multispin quintet states and emission in dimers and trimers comprising multiple diphenylhexatriene using the transient EPR and ODMR methods. The authors now are focusing on *room-temperature emission mediated by quintet states as the optical spin readout*.

Although the present manuscripts are improved comparing to the previous submission on the singlet-fission study of the oligomer sample, some explanations interpreting the data are highly suspicious and odd, as responded to the author comments below:

1) Fig.2: From c), the authors mention that the eaacea-pattern triplet formation is due to the reduction in the exchange interaction to the very weakly triplet pairs with selective formation of $m_s = 0$ in the individual triplets. However, the free triplet in the selective formation in $m_s = 0$ might be a result of the triplet-TTA process with $3(T1+T1) \rightarrow 3(TT) \rightarrow T+S0$ (DOI: 10.1073/pnas.1820932116) because the separated quintet T+T character (Q0) can undergo the Q-T conversion to $3(T+T)$ at $m_s=0$. Please comment on the possibility of the participation of the TTA product in the present systems especially in the trimer samples. If no triplet-TTA product is conclusive, why this TTA does not occur? What is the bottleneck of this?

The reviewer raises a good question. The possibility of TTA involving a $3(TT)$ intermediate

is indeed relevant in certain systems, particularly when free triplets are mobile and able to reencounter each other. However, based on several experimental observations, we find no evidence for TTA as a dominant pathway at room temperature in our DPH-based oligomers:

- TrPL measurements of all three oligomers show comparable dynamics at room temperature (Fig. S6), including the dimer which generates only one triplet per excitation (via ISC) and thus cannot support TTA. The comparable kinetics indicate that the delayed emission dynamics are not governed by annihilation processes from $T1 + T1$ in the trimers, as we would expect diverent emission features compared to the dimer, especially at room temperature.

-> Reviewer comment: The second and the third decay components in Fig. S6 can be explained by the delayed fluorescence via T+T causing singlet-TTA. Even in the dimer system, equilibrium among S1 and TT and T1+T1 will induce the triple exponential decay as reported previously (J. Phys. Chem. C 2021, 125, 33, 18287–18296). Such very weakly

coupled T1+T1 should induce the superposition of the singlet, triplet and quintet characters in the presence of the small J-coupling as shown in the above JPC2021 paper. This will thus cause the 3T...T population during the equilibrium among S1 and TT and T1+T1, which means that Fig. 5 is totally odd. Thus, the occurrence of the singlet-TTA does not exclude the triplet-TTA contribution for generating the isolated triplet product showing the eaacea-pattern.

- Room-temperature ODMR shows no triplet signal, only quintet emission, confirming that triplets, or intermediate 3TT states, do not participate in emission under ambient conditions. The triplet signal in ODMR at 80K is further discussed in question 4.

-> Reviewer comment: This is not the reason for the absence of the triplet TTA at all, because the triplet TTA does not contribute to the emissive S1 generation but to the dark individual T1 product.

- Time-resolved magnetic field-dependent PL (magPL) shows no signature of a reversed magnetic field effect, which is typically associated with TTA-dominated emission (e.g., negative MFEs at late times due to reverse SF).

-> Reviewer comment: The absence of the MFE does not exclude the triplet TTA scenario, when the exchange coupling between the 3(TT) and 1(TT) characters is much larger (or much smaller) than the applied magnetic field.

- Transient absorption reveals a long-lived triplet plateau persisting to hundreds of microseconds. However, the integrated PL (Fig. 2c) shows that all emission is complete by $\sim 2 \mu\text{s}$, with no delayed fluorescence corresponding to this residual triplet population ($\sim 30\%$ signal) in TA. Even if TTA were occurring at later times, a fraction of events would be expected to repopulate S1 + S0 and result in observable PL, which cannot be observed on these timescales.

-> Reviewer comment: Given that the plateau persists hundreds of microseconds and that the no delayed emission is observed, it is very likely that $3(T1+T1) \rightarrow 3(TT) \rightarrow T+S0$ is taking place in the dimer and in the oligomer systems during the equilibrium. Again, I note that the triplet TTA is not the process to emit the delayed fluorescence but to produce the dark isolated T1 species. Overall, I strongly recommend that the authors rewrite the manuscript accordingly. Nevertheless, the occurrence of the triplet TTA does not exclude the quintet-(and T+T) mediated emission in the present study, as detailed above.

Taken together, these data do not support TTA being a contribution our systems. Instead,

the $m_s = 0$ triplet polarization pattern is best assigned to forward singlet fission, where weakly exchange-coupled states dissociate into free triplets. We acknowledge that a minor contribution from TTA cannot be entirely excluded at low temperatures where triplet motion is more restricted and decay is slower, as further discussed in question 4. We kindly refer to the potential involvement of a 3TT state also to question 4. We have added a discussion about the negligible TTA at room temperature, but potential influence at low temperatures, to the Supporting Information, Section 6.

Finally, we note that there are intrinsic bottlenecks that likely suppress TTA in our system. Although the DPH oligomers are not fully rigid and can undergo torsional motion around single bonds, the triplet excitons remain largely localized on individual chromophore units. This localization limits the spatial overlap between excitons, reducing the probability of encounter and annihilation. Furthermore, while some rotational flexibility exists, the covalent connection of chromophores in a defined oligomeric architecture prevents the type of long-range exciton diffusion that is typically required for efficient TTA, as observed in disordered films or crystalline systems with mobile triplets. These structural features make mutual triplet encounters unlikely and thus further diminish the potential for TTA.

-> Reviewer comment: I would agree that such flexibility would cause the TTA for the luminescence from the T_1+T_1 to the S_1 state. However, the occurrences of the singlet-TTA and the quintet-mediated TTA does not exclude the triplet-TTA to generate the long-lived plateau for the isolated triplet product.

2) The weakly SCTP spin polarization model was originally reported in 2020 (doi.org/10.1039/C9SC04949E) which is also not cited here. In this, the eaacea-pattern was explained by the sequential spin polarization transfer of the S-Q0 coherence in the secondary TT pair to the separated T+T state in the presence of the negative J-coupling. Please describe this possibility for the interpretation of the present spectra in the trimer systems. This reviewer also believes that the reversed polarization is obtained in the separated triplets when the D value is positive. However, the eaacea-pattern can be explained by the separation of the Q0 character to the individual triplets when D is negative.

We thank the reviewer for pointing out the original report of the SCTP model (Ref. 31 in the revised version). In our manuscript, we used the term “spin-correlated triplet pair”, analogously to the “spin-correlated radical pair” concept widely applied in organic

photovoltaic, to denote a pair of weakly coupled triplets with non-equilibrium spin populations. In agreement in Weiss et al. (Nature Physics 2017), the wavefunctions of a SCTP (or weakly-coupled pair) are analogous to spin physics of a SCRP but with triplet pairs instead of spin pairs. We did not initially intend to refer specifically to the electron spin polarization transfer (ESPT) model as proposed in the cited work, but we recognize that the SCTP model described there is highly relevant, and we now cite and discuss it in the revised manuscript as well as changing our terminology throughout the manuscript. Please refer to the next question for a more detailed discussion about the spin-mixing mechanism.

We note that in our system, the exchange coupling J is assumed to be positive, as further discussed in question 6, placing the $5TT$ state lowest and $1TT$ highest in energy. However, as the ESPT model depends on transient low- J conditions rather than strictly negative J , a $Q0$ -originated spin correlation can still be preserved and transferred to the dissociated $T + T$ pair. Furthermore, our manuscript gives the absolute values $|DQ|$, whilst the exact simulations use indeed use negative D values.

-> Reviewer comment: OK.

3) In the dimer sample, the ISC triplet is claimed to be generated, suggesting that the SF efficiency is low. Is there any evidence of the ISC competing with the SF from the transient absorption data? Was the initial quantum yield of the triplet formation too far less than 200 % by the absorbance at the nanosecond regime before the triplet decay? Please obtain the ISC rate of the monomer sample and discuss the SF efficiency by comparing the SF and ISC rate constants from the transient absorptions. In addition, the electron spin polarization transfer of a quintet TT to a conformation changed TT can cause several electron spin polarization patterns depending on the dipolar orientations in the separated triplets with the quintet character as described in Ref. 43.

We thank the reviewer for highlighting this point and apologize if our use of terminology was misleading. In the manuscript, we refer to triplet formation via SF as the formation of free $T_1 + T_1$ excitons. However, in all oligomers, we observe a fast initial SF process with

a rate of $k^{\text{SF}} = 2.8 \times 10^{10} \text{ s}^{-1}$ (see Fig. S.15) by formation of the intermediate $^1(TT)$ state. The subsequent formation of T_1 states differs across the oligomers. In the dimer, the SF process does not proceed to fully separated triplets. Instead, it results in either strongly exchange-coupled triplet pairs $5(TT)$ or triplet formation via ISC from $1(TT)$ to T_1 . Thus,

the

initial SF process from S1 to 1(TT) does not stay in competition with ISC from S1, as the 'initial' SF process happens on the ps timescale. In the dimer, and partly in the trimer (DPH)₃, we observe a dynamic equilibrium between 1(TT), 5(TT), and weakly exchange-coupled

(T...T) states. Under these conditions, ISC can occur eventually from 1(TT) to T1.

We are aware that there are different terminologies in literature and we have revised the description in the new version to better reflect this complexity.

We are grateful for the insightful comment regarding the spin-mixing mechanism. We carefully reviewed Chem. Sci. 2020 (now Ref. 31) and J. Phys. Chem. B 2020 (now Ref. 32) and included a discussion in our manuscript, line 138 - 150. In our analysis, we consider both proposed mechanisms for quintet dissociation: the low-J coherent mixing regime and the stochastic/geometry-driven mechanism involving conformational changes. Based on the spin polarization patterns observed in both quintet and triplet signals, which align with expectations from the low-J regime, we find our data more consistent with coherent singlet-quintet mixing in a regime of small exchange interaction.

While we cannot fully exclude stochastic or geometry-driven quintet dissociation pathways, the observed polarisation signatures point toward a low-J window enabling mixing before eventual dissociation into T1 + T1. We now include a discussion in the main

text addressing both mechanistic frameworks, and a more detailed explanation in section 4 of the Supporting Information. We updated the main text as follows:

"Several models have been proposed to describe how quintet state 5TT emerges from an initially formed S = 0 triplet pair 1TT. In the low-J regime, where the exchange interaction J(t) transiently becomes small and comparable to the zero-field splitting D, coherent mixing between singlet-quintet mixing can occur on nanosecond timescales.^{5,16} A subsequent increase in J projects the mixed state onto 5TT, resulting in spin-polarized quintets. In contrast, another mechanism discussed in literature includes J remaining large but fluctuates due to structural or vibronic motions, which can enable anisotropic spin-relaxation pathways into the quintet manifold.^{31,32} These include spin polarization

transfer from 1TT or 5TT to weakly coupled T+T states, modulated by torsional conformations and dynamic J-coupling.³² While our data with dominant $m_s = 0$ polarization and absence of ± 2 signatures aligns more closely with the low-J scenario¹⁶, the flexibility of our oligomers may transiently allow access to such fluctuating-J configurations. A detailed discussion and comparison of these pathways is provided in Section 4 of the Supporting Information”

Whilst we could focus even further on possible mixing mechanisms, we are reluctant to make the manuscript any longer since it would detract from the central focus of our work: to establish a comprehensive structure-function relationship across DPH oligomers for triplet formation via SF or competing ISC pathways as well as to bring attention to optically addressable high-spin states.

-> Reviewer comment: OK.

4) Regime II and III (Page 11): This is a most interesting regime in the present study but is

controversial in the present version. First, please explain why the $3(T\dots T)$ is omitted in the spin state mixing in the SCTP. For understanding the triplet feature in the ODMR spectra, this $3(T\dots T)$ can be a strong candidate because of highly possible equilibrium of $1(S1S0) \rightleftharpoons 1(T\dots T) \rightleftharpoons 5(T\dots T) \rightleftharpoons 3(T\dots T)$ in the weak SCTP to contribute to sublevel transitions in the $3(T\dots T)$ for the change in the PL intensity. But this reviewer did not find

any description about the sources of the triplet resonance in the ODMR spectrum.

We thank the reviewer for giving the possibility to clarify the role of $3(TT)$ states in our systems. As shown in the reviewer’s previous reference (PNAS 2018, now Ref. 37), a proposed mechanism in SF is the population of $3(TT)$ via spin relaxation from $1(TT)$ which could lead to $3(TT) \rightarrow T1$ transitions, which we have considered before as well. We have carefully reconsidered this model in the context of the new magnetic resonance measurements, and especially the ODMR data. The ODMR results do not support a major role for the $3(TT)$ or $3(T\dots T)$ state in our systems.

Specifically, we would expect any thermally accessible $3(TT)$ or $3(T\dots T)$ state to be present across all oligomers which enter the weakly-coupled triplet pair, including the dimer. However:

- The dimer enters the weakly-coupled regime $n(T\dots T)$ but its ODMR reveals only quintet signals in the delayed photoluminescence. If a possible equilibrium of $1(S1S0) \rightleftharpoons 1(T\dots T) \rightleftharpoons 5(T\dots T) \rightleftharpoons 3(T\dots T)$ or $5(TT) \rightarrow 3(TT) \text{ @ } T1$ pathways were active, one

would expect corresponding triplet signatures in ODMR of the dimer. However, we do not observe a triplet signal in the ODMR of the dimer or at room temperature for the trimer. The dimer only shows a triplet signal in trEPR that is consistent with direct ISC to T1. If this signal would have the equilibrium with $3(TT)$ or $3(T\dots T)$ states as origin, it would have to be present in Me-(DPH)₃ as well. However, the triplet signals for (DPH)₂ and Me-(DPH)₃ have spin polarization from different processes (SF vs ISC), why they cannot origin from the same state.

- Triplet features in ODMR on the trimers are only seen at low temperatures (≤ 80

K). If a $3(TT)$ -mediated pathway were dominant, its contribution would likely increase with temperature, due to faster spin-lattice relaxation and better accessibility, which we do not observe. Room temperature ODMR only shows a quintet contribution to the PL. This is supported by the similar trPL at room temperature of both trimers and the dimer.

These observations, especially the additional ODMR spectra, suggest that $3(TT)$ or $3(T\dots T)$

do not contribute significantly as an intermediate state in our systems. We now discuss this in the main text, lines 225 – 245.

The reviewer is right that the origin of the triplet signal in the ODMR was not sufficiently discussed. We observe a triplet contribution to the ODMR spectrum of both trimers at 80 K, but not at room temperature. As Me-(DPH)₃ shows no evidence of ISC in EPR, and the dimer (DPH)₂ – where triplets form solely via ISC – shows no ODMR triplet signature, we assign this signal to SF-derived triplets that remain partially localized at low temperature.

Their weak coupling and extended lifetime allow them to contribute to ODMR at cryogenic conditions. At higher temperatures, delocalization is more efficient and relaxation is faster, why SF-derived triplet states delocalize and decay to the ground state and only $5(TT)$ quintet emission dominates. This interpretation is consistent with the

temperature-dependent trPL, showing a slower decay at lower temperature, whilst trPL at room temperature is similar for all three oligomers, proving quintet contribution as the major high-spin state contribution. We added this ODMR discussion together with the TTA discussion at room temperature to Section 6 of the Supporting Information.

-> Reviewer comment: Although it would be better that the authors add the ODMR discussion, the authors do not mention the possible origin of the triplet-resonance of the ODMR at low temperature including why the triplet resonance is absent at higher temperature and in the oligomer sample. I do not think that the triplet contribution is minor at lower temperature in the ODMR spectra.

5) The authors claim that the quintet is the major source of the delayed fluorescence from Fig. S7. But this reviewer disagrees with this, because singlet TT (and triplet TT) seems to be stronger origins of the delayed fluorescence in Fig. 5. Rewriting the relevant sentences should be considered.

We thank the reviewer for pointing out this ambiguity. We agree that the early-time delayed fluorescence (within the first ~100 ns) is dominated by recombination from singlet $1(TT)$ states, as also evident from the TA and (magnetic-field) PL dynamics. Our intention was to highlight that the dominant contribution of high-spin states (i.e. states beyond singlet states) to delayed emission originates mainly from quintet states.

We have now revised the relevant sentences to clearly distinguish between total delayed fluorescence (which includes singlet/ $1TT$ emission) and the contribution from high-spin states (where the quintet dominates), as our study focuses on these high-spin pathways and their optical accessibility.

-> Reviewer comment: OK.

6) From the temperature dependence of the ODMR spectra (Fig. 4c), the triplet contribution is predominant at lower temperature. But there is no explanation on the source of the triplet resonance that contributes to the delayed fluorescence. Given the negative J-coupling in the strongly coupled TT, $3(TT)$ state should be populated rather than the $5(TT)$ at lower temperature in Fig. 5. Thus, I recommend to the authors to reconsider the origin of the ODMR including the effect of the equilibrium between the strongly coupled $1(TT)$, $3(TT)$ and $5(TT)$ which are not drawn in Fig.5.

As described in our response to question 4, we have considered the equilibrium between the singlet, triplet, and quintet TT states and its potential relevance for ODMR. However,

our experimental data – particularly the absence of ODMR triplet signals in the dimer and the appearance of triplet features in the trimers only at low temperatures – support a model where $5(TT)$ is the dominant high-spin contributor to the delayed fluorescence, and $3(TT)$ plays a minimal role. We have clarified this interpretation in the main text, line 225 –245 and added a discussion about the triplet contribution of ODMR in Supporting Information.

We also note that our data supports a positive exchange interaction, where the quintet state $5(TT)$ lies below the singlet $1(TT)$. This is supported by the temperature-dependent ODMR, where the $5TT$ signal has an additional temperature-activation in ODMR with increasing temperature from 30K, whilst the trEPR shows no temperature activation in the quintet signal. This suggests that, whilst $5(TT)$ states are always accessible, even at cryogenic temperatures, the $5(TT)$ state thermally access the $1(TT)$ state on the back pathway. This is in combination with the overall temperature activation from $1(TT)$ to $S1$, which is visible until 30K (Fig. 3b) and supported by temperature-dependent TA (Fig. 4c). The additional rise of quintet signal until 80K stays in agreement with an exchange energy of about 1-2 meV. We have clarified this point in the revised manuscript (line 318-320) and updated Figure 5 to reflect the energy-level ordering and thermally activated transitions accordingly.

-> Reviewer comment: Again, the authors still do not mention the possible origin of the triplet-resonance of the ODMR at low temperature including why the triplet resonance is absent at higher temperature and in the oligomer sample. I am not asking why the quintet TT is contributing to the ODMR, but why and what triplet-resonance is contributing to the fluorescence. First, please clarify whether the intermolecular triplet-triplet annihilation is associated with the ODMR (as in the paper in DOI: 10.1002/adfm.202212640). Please also rationalize why the $3(TT)$ and $3(T...T)$ are not accessible to $1(TT)$ which may probably be accessible to the fluorescent state at lower temperature, although the above paper says that the intermolecular triplet-pair can access to the fluorescent state. It is very important to describe what is the most likely origin of the triplet-resonance of the ODMR spectrum in the Nature Comm journal for the quality and clarity of the paper for the broad readership.

Overall, this reviewer sees that the present interpretations of the whole data are too

preliminary. In particular, they omit the contributions of the 3TT being highly odd considering the previous reports (for example, DOI: 1038/s42004-018-0008-0 in DPH samples). Thus, the present manuscript is not worthy of the publications in the Nature Chemistry.

We understand the concern about the potential role of 3(TT) in light of previous reports, including our own earlier study, which proposed a 3(TT)-mediated mechanism based solely on optical data. However, in the present work, the inclusion of magnetic resonance measurements – particularly temperature-dependent ODMR and trEPR – provides direct spin-resolved evidence that the 3(TT) state does not play a significant role in the singlet fission process or delayed fluorescence in our oligomer systems. As outlined in our responses to points 2, 4, and 6, we now discuss this interpretation in more detail in both the main manuscript and the Supporting Information.

-> Reviewer comment: Overall, this reviewer still sees again that the present interpretations of the whole data are too preliminary, although they are trying to focus on the quintet ODMR results and responded to the comments from reviewers 2 and 3. In particular, they omit the contributions of the 3TT and 3T...T states (See Fig.5) which are highly odd considering the previous reports (for example, DOI: 1038/s42004-018-0008-0 in DPH samples). Thus, the present manuscript is not worthy of the publications in the Nature Comm.

Reviewer #3 (Comments for the Author):

Greenham and co-workers' manuscript describes the study of three oligomeric [two trimers and one dimer] diphenylhexatrienes. The authors previously introduced this chromophore as an alternative to classical singlet fission (SF) materials because their singlet and triplet energy levels are consistent with such a process. For this study, they mainly studied the more rigid trimer Me-(DPH)₃ and used the other two oligomers to try to elucidate the photochemistry of these molecules. They find that, in this small series, it is possible to observe triplet formation via both intersystem crossing (ISC) and SF, as well as the formation of strongly coupled triplets showing quintet character. Using transient electronic absorption (TA), [transient] electron paramagnetic resonance EPR [trEPR], and optically detected magnetic resonance (ODMR) they explore the differences in relative populations of SF triplets, ISC triplets, and quintet states as well as the (delayed) photoluminescence (PL) of these materials. The authors propose a

tentative pathway for the generation of the high-spin states and conclude that most of the delayed PL that is observed originates from the quintet state. They also observed, by ODMR, that high-spin PL at room temperature can be mainly attributed to the quintet state.

The reported findings are interesting but, in the opinion of this reviewer, not suitable for publication in Nature Chemistry for the following reasons:

I) Considering their earlier published papers [JACS 2024, ref. 23 and JACS 2023, ref. 30],

the work presented here appears to be rather incremental. In particular, two out of the three molecules presented in Figure 1 were reported and extensively analyzed using optical spectroscopy. I understand that most of the analysis in this manuscript is based on magnetic resonance techniques, but it is nonetheless true that the data presented here would have fitted perfectly in that manuscript and resulted in a more complete study. This is even more true if one considers that some of the conclusions regarding the high-spin species in their JACS 2024 appear to be in conflict with the ones presented here. For instance, in their JACS they assign the “plateau” in Fig. 1c as

“Isolated T1” [Fig. 4b in JACS 2024] because “one of the two triplets initially produced by SF has already been lost” [Supplementary Information from JACS 2024, p. 7].

I cannot imagine that the authors did not know the results presented here when that paper was submitted and subsequently accepted [received on July 31st and published on October 17th, 2024].

II) It is hard to see how the message of this manuscript is sufficiently general to merit publication in the top journal of chemistry. All three molecules give different, a priori not predictable, high-spin dynamics. I fail to see how the results presented here allow any level of molecular design we did not have previously [e.g. their motivation to try the double methylation]. In addition, it refers exclusively to this family of chromophores and we have no information regarding any possible implication on other systems. This is even more true if one considers that the results of these measurements are strongly dependent on the molecular conformations, from which we have only limited information in the current version of the manuscript. The influence of the double methylation on the electronic properties of the chromophore is also not discussed at all.

III) The results described in this manuscript read as a very specific study on the photochemical properties of a new, more rigid version of a previously reported diphenylhexatriene trimer, which is more suitable for a specialized journal rather than a general chemistry journal.

We acknowledge the concerns regarding novelty and generality. These points were important to address, and as outlined in our responses to Reviewer 1, we have revised the manuscript to more clearly articulate the unique contributions of this work.

Specifically, the relationship to our prior study (JACS 2024, ref. 25) is now explicitly discussed in the revised manuscript. That study was based solely on optical data and proposed a model involving $3(TT)$ states. The current work adds a new dimension through magnetic resonance techniques – particularly room-temperature ODMR – which directly resolve the spin multiplicity of the emitting state and show that delayed fluorescence arises from quintet states. This represents a revised mechanistic understanding, made possible by new experimental evidence.

Clearly, this updated interpretation is of interest and not beyond debate, as also reflected in Reviewer 2's detailed questions about whether $3(TT)$ states might still play a role. We now discuss this issue explicitly in the revised manuscript, including both mechanistic possibilities and their implications for the spin dynamics. We hope this provides a clearer and more comprehensive account of the system.

Furthermore, this work focusses on a new methylated DPH trimer, Me-(DPH)₃, which we

now fully characterize. This molecule shows exclusive SF-mediated triplet formation, with ISC entirely suppressed, offering a structurally tuneable platform for high-spin state engineering. The ability to selectively suppress ISC through minimal structural changes, while maintaining SF efficiency and enabling optical quintet emission at room temperature, provides a valuable design principle for future materials targeting spin-based optoelectronics.

Prior to submission to a more suitable journal, I would recommend the authors address some of the following [not comprehensive] remarks:

a) One of the main assumptions for studying this series is that the 1,6 dimethyl diphenylhexatriene chromophore shares the same electronic and magnetic properties of diphenylhexatriene. While both compounds will certainly be close, this reviewer is concerned that given sufficient differences, the system will not behave as a direct

comparison to (DPH)₃, but rather as a more complex DPH–MeDPH–DPH. This is why the basic characterization [absorption, fluorescence, PLQY, and fluorescent lifetimes] of any new compounds is of critical importance before engaging in more advanced characterization. In this case, I would argue that a detailed analysis of the TA data as compared to the DPH would be critical.

We thank the reviewer for raising the point regarding the assumption of comparable electronic structure between the Me-substituted DPH trimer Me-(DPH)₃ and the unsubstituted compound (DPH)₃. To address this, we have performed a direct comparison of the steady-state absorption and PL spectra, TCSPC and TA of Me-(DPH)₃, the monomeric DPH, and the unsubstituted trimer (DPH)₃, as shown in Figure R1. Both absorption and emission spectra of the Me-substituted trimer closely resemble those of the monomer, and are nearly indistinguishable from the unsubstituted (DPH)₃ trimer (Fig. R1 a). The vibronic structure is well preserved, with a small reduction of vibronic features and minor broadening, which is consistent with earlier reports of (DPH)₃. Also, no significant spectral shifts or additional low-energy shoulders are observed, suggesting that the chromophore units in Me-(DPH)₃ maintain their individual electronic character without significant conjugative or electronic perturbation from neighbouring units. This comparison confirms that the introduction of methyl groups does not substantially alter the optical properties relative to the previously studied (DPH)₃. This behaviour supports the assumption that Me-(DPH)₃ retains the essential electronic features of DPH and can be used as a model for investigating spin and excited-state dynamics intrinsic to the DPH core.

We also compared the excited-state dynamics of Me-(DPH)₃, (DPH)₃, and the monomer using TCSPC (Fig R1 b). The monomer shows a lifetime of approximately $\tau = 4.1$ ns, consistent with previously reported values for DPH derivatives and reflecting typical S₁ decay dynamics. The trimers, in contrast, display multiexponential decays with longer-lived components. This behaviour is characteristic of systems undergoing intramolecular singlet fission, where the emissive S₁ state is coupled to a triplet-pair manifold. The delayed fluorescence observed in both trimers arises from slow re-fusion of weakly coupled triplet pairs, and including both 1(TT) and 5(TT), back to S₁, which is absent in the monomer. Importantly, the Me-substituted and unsubstituted trimers

exhibit comparable excited-state dynamics, further indicating that methyl substitution does not disrupt the fundamental photophysics of the system.

To further assess the nature of triplet formation in these systems, we compared the nanosecond TA spectra of Me-(DPH)₃, and (DPH)₃. As the monomer has no pronounced triplet feature, we show it in comparison with (DPH)₂ which enters the TT manifold but has no triplets by singlet fission, instead one triplet per photoexcitation by ISC. This comparison allows us to differentiate between triplet populations arising from singlet fission or ISC.

The dimer (DPH)₂, which does not produce triplet multiplication by SF, shows only a small triplet signal plateau, indicative of an inefficient ISC channel. In contrast, both trimers, Me-(DPH)₃ and (DPH)₃, show higher formation of long-lived triplet features, consistent with triplet-pair generation through SF. The PLQY is in accordance with the TA

data with $F = 73\%$, $F = 31\%$, and $F = 21\%$ for (DPH)₂, Me-(DPH)₃ and (DPH)₃, respectively.

Importantly, while (DPH)₃ displays a higher final triplet plateau than Me-(DPH)₃, this signal includes contributions from both SF and ISC-derived triplets. The Me-substituted trimer, however, exhibits triplet formation solely via singlet fission, with no observable ISC contribution. This distinction confirms that Me-(DPH)₃ serves as a cleaner model system for studying SF-specific dynamics without the complication of parallel ISC pathways - aligning more directly with our design goal of isolating pure SF dynamics. The origin of this difference is likely structural. While a full mechanistic study lies beyond the scope of this work and would require additional synthetic and computational investigations, we suspect that steric hindrance introduced by the methyl substituents in Me-(DPH)₃ influences the torsional flexibility and chromophore coupling. This could suppress conjugation and through-bond communication, thereby inhibiting ISC and favouring SF pathways. This reasoning would also be consistent with prior structure-function correlations in SF systems. We highlight this as a promising path for further investigation, particularly as it connects molecular design with controlling spin dynamics.

As the basic characterization of these systems has been discussed in previous publications and they share similar electronic properties, we summarize the key results in the main text and provide the full dataset and analysis in Section 3 the Supporting

Information.

b) The authors might need to add more information to the main text and supplementary information. For instance, they do not report absorption spectra of the compounds nor fluorescent quantum yields or fluorescence lifetimes. The TA data is only presented as main figures, in which the wavelength range is likely reduced with respect to the original data and, thus, an additional typical feature centered around 725 nm is not visible. Did the authors use a different setup? It is not possible to know since that information is not available in the supplementary information of their JACS 2024. The authors also do not report the full set of simulation parameters [e.g. for their trEPR spectra] which makes it impossible to reproduce or verify their data analysis procedures.

We are happy to add a more detailed experimental information. We have now included the full optical characterization of Me-(DPH)₃, including absorption spectra, fluorescence quantum yields, and fluorescence lifetimes, in the Supporting Information, Section 3, and a short discussion in the main text. This is in addition to the optical data for (DPH)₃ previously reported in our JACS 2024 publication, which are now also consolidated in the updated Supporting Information for clarity and completeness.

Regarding the TA measurements, the data presented in the manuscript were acquired using a modified setup configuration, optimized for probing short-wavelength TT-related features with high signal-to-noise ratios. This spectral tuning towards the blue region was implemented intentionally to reduce the required excitation fluence and mitigate potential sample degradation (as visible in DT/T in 10⁻³ order of magnitude, see Fig. 4)

—
critical for reliably tracking long-lived TT dynamics. As a result, the spectral window does

not extend to the 725 nm region. We have clarified our probe range in the Methods section.

For the trEPR and ODMR simulations, we have provided the full set of simulation parameters in Tables S1 – S3 of the Supporting Information, Section 2. These include all spin Hamiltonian parameters and fitting conditions used to reproduce the data shown in the main figures.

We trust that these clarifications and additions address the reviewer's concerns and provide the level of detail necessary for full scientific rigor and reproducibility.

In any case, the manuscript lacks information about the conditions under which the

experiments were carried out. For instance, despite reading the text several times, I was unable to know with certitude if any given experiment was carried out in a dilute solution or a dispersed film. The authors add a line to the sample preparation stating that both preparations give the “same signals,” which is a statement I find very confusing as it is most improbable to observe the same spectra under these two different sample preparations. Such a statement should most definitely be comprehensively proven.

We thank the reviewer for pointing out the need for more clarity regarding the experimental conditions. In response, we have now explicitly detailed the measurement conditions for each experiment in the revised Methods section of the manuscript and Supplementary Information and added comparison measurements of solution and films (see Fig. R2).

- Magnetic resonance experiments were performed either on frozen toluene solution samples (<170 K, i.e. melting point of toluene) or polystyrene films (>170 K). This is due to instrumental constraints and the need to maintain molecular rigidity during detection of spin-polarized states. Fig. R2 shows that trEPR spectra in frozen solution and films exhibit the same spectral features, differences in signal-to-noise arise from the lower laser power used for films to avoid photodegradation, which occurs more likely in the solid state.
- Cryogenic optical measurements (e.g., TA, magPL) were performed on polystyrene films, as required by the experimental setup: most optical cryostats operate via a cold finger to ensure optical transmission and therefore only accommodates solid-state samples. Figure R2 presents a direct comparison between room-temperature measurements in solution and in film form. The data show similar kinetic behaviour in the relevant time window (nanoseconds to tens of microseconds), with only a minor divergence in plateau amplitude, which we attribute to variations in optical density and scattering between sample formats. For film preparation, a carefully optimized concentration of 0.1 wt% was used to exclude intermolecular interactions. Concentrations above 1 wt% show aggregation artifacts (e.g., spectral broadening and an additional central peak), which were strictly avoided. We hope this clarification addresses the reviewer’s concern and reinforces the

robustness of our experimental observations across sample formats.

c) While the manuscript is described as focusing on a single molecule Me(DPH)₃, two other compounds have also been studied to enable a better interpretation of the data. However, this is not the case for all the analyses that were made and are notably missing in cases where a different molecule would have been [arguably] more suitable, e.g., the variable temperature trEPR and ODMR.

While the manuscript indeed focuses on Me-(DPH)₃, the compounds (DPH)₂ and (DPH)₃

were studied in parallel to support the mechanistic interpretation. For temperature-dependent trEPR and ODMR measurements, we deliberately focused on Me-(DPH)₃ because it exhibits singlet fission as the sole triplet formation pathway, without competing ISC. This enables unambiguous interpretation of the temperature dependence, isolating effects intrinsic to the SF mechanism and avoiding convolution with ISC-related processes, which could introduce separate temperature-activated behaviour.

Furthermore, since our interest lies in probing the energetics of the correlated triplet-pair manifold and the activation barrier to reformation of S₁, Me-(DPH)₃ serves as the most appropriate model. However, to demonstrate that SF remains operative even at cryogenic temperatures in the (DPH)₃ system, we have now included the trEPR spectrum of (DPH)₃ at 10K in the Supplementary Information (Fig. S10) and refer to it in the temperature discussion about Me-(DPH)₃. The temperature range selection and its relevance are discussed in the following question response.

d) Regarding the variable temperature trEPR, I would argue that making this measurement for all three compounds [and more temperatures] would have been informative. In that way, the conclusion of the evolution of the contributions to the triplet and quintet states would be better justified. In its current form, it is based on measurements at only three temperatures and, I guess, the qualitative analysis of the shape of the spectra because no corresponding simulations or data analysis is presented.

We thank the reviewer for this suggestion. In response, we have now added simulations for the full set of available trEPR spectra at different temperatures to the Supporting Information (Figure S9 in Section 1 and Table S2 in Section 2) and referred to it in the

main

text. These simulations highlight that at lower temperatures, an additional signal by ISC begins to contribute to the observed signals, consistent with our assignment of competing pathways in these systems.

Regarding the number of temperatures measured: the temperature range for trEPR is inherently limited by the spin dynamics of the system. Above ~ 100 K, spin-lattice relaxation and decoherence become too fast to resolve the transient spin polarization signals. This is consistent with our pulsed EPR measurements, which show a T2 time of $1.7 \mu\text{s}$ at 20 K, typical for organic materials, and indicating that coherence times fall below detection limits at higher temperatures. ODMR in contrast is based on different measurement configuration, including continuous light excitation and more sensitive optical detection, which makes it possible to measure up to room temperature.

The measured temperature window (10-100 K) still captures the relevant thermally activated regime for the reverse singlet fission process (from $1(TT)$ to $S1$), as evidenced by

correlated magnetic resonance and PL dynamics. We thus believe that this range provides a meaningful insight into the activation behaviour of high-spin states.

As noted in the previous response, we focused the temperature-dependent analysis on Me-(DPH)₃, as it cleanly exhibits triplets generated exclusively via singlet fission, ensuring that the observed temperature effects can be unambiguously attributed to the SF pathway rather than ISC.

e) They observe that most of the detected PL is emitted by $2 \mu\text{s}$, which roughly coincides with the decay of the quintet state by trEPR. They interpret this observation as evidence that quintet states are the predominant contributor to PL. However, they present no data on fluorescence quantum yields and never specify if they detect any phosphorescence [which would be at the edge or outside of the detection limit of the camera and in the region of low QE]. I would argue that this conclusion is not currently sufficiently supported by the data they present.

We appreciate this comment, and have clarified our explanation and highlighted the evidence more in the revised manuscript.

The main experimental evidence for quintet-mediated PL at room temperature is derived from ODMR. The ODMR spectrum shown in Figure 3c exclusively exhibits quintet signatures at room temperature. If triplet states contributed to the PL, triplet ODMR

signals would be present in this very sensitive technique. To make this argument clearer, we now include a dedicated panel highlighting the room-temperature ODMR data in the revised version.

Furthermore, integrated PL measurements demonstrate that the emission is nearly complete by 2 μ s, while TA data reveal that triplet populations persist for several hundred microseconds. This discrepancy strongly suggests that triplet states do not contribute significantly to the observed PL, reinforcing the conclusion that quintets dominate the emission process.

Regarding phosphorescence, we do not observe any emission in the expected near-infrared region (\sim 830 nm for a 1.5 eV triplet). Our detection range extends to \sim 850-900 nm, which would allow observation of at least the onset if present. However, given the relatively low triplet population (as most excited states return via S1 or quintet emission) and the inherently weak nature of phosphorescence due to the energy gap law in this wavelength range, no detectable signal is observed – even with the use of a sensitive gated ICCD detector at 30 K.

As also addressed in Reviewer 2, question 4, the triplet ODMR spectrum becomes visible only at cryogenic temperatures, and not at room temperature. While phosphorescence could, in principle, yield an ODMR signal if the different triplet sublevel yields different strength of spin-orbit coupling to the ground state, we observe no phosphorescence under any conditions. We therefore attribute the low-temperature triplet ODMR signal to residual TTA, as discussed in detail in the above question and in the revised Supporting Information, Section 6.

Minor (or major?!) concern:

Of the compounds depicted in Fig. 1, two [(DPH)₂ and (DPH)₃] were already reported in

their recent JACS 2024 [received in July 31th and published in Oct. 17th]. That is not so remarkable as they also referenced this work in ref. 23. What is not normal, however, is that, in the synthetic procedures, they do not mention that these compounds are already reported. In fact, in three out of six reported procedures, the synthetic procedures are an exact copy-paste from their JACS 2024 with one difference: in the present manuscript they removed one sentence “(synthesized as previously reported)”.

The reference to N in their JACS 2024 refers to their earlier work published in JACS 2023

and Chem. Sci. 2023. I do not find this way of proceeding acceptable, but I can grant them the benefit of the doubt. The reported NMR spectra in this manuscript are also exactly the same as those reported in their JACS 2024, but this time, they did not add the high-resolution mass analysis to the supplementary information.

We thank the reviewer for this important observation. As correctly noted, the compounds (DPH)2 and (DPH)3 were previously reported in our JACS 2024 publication, and we acknowledge that this was also cited in the current manuscript. Our intention in including these compounds again was to facilitate a clear and direct comparison with the newly introduced Me-(DPH)3 derivative.

To clarify, all three compounds were synthesized in same way as in the previous reports in our laboratory. Thus, as the synthetic procedures for (DPH)2 and (DPH)3 are unchanged

from our earlier report, we used the same protocols. We reprinted the NMR and synthetic procedure in the Supporting Information for direct comparison purposes. We appreciate the reviewer's point that this should have been more explicitly stated in the Supporting Information. We have now revised the Supporting Information to clearly indicate that the procedures and NMR spectra for (DPH)2 and (DPH)3 are reproduced from our previous publication (JACS 2024) for reference and comparison purposes, and only the newly introduced Me-(DPH)3 is added.

We hope this clarifies our rationale and trust the updated labelling and attribution now meet the standards of transparency and good scientific practice.

Comments regarding manuscript NCOMMS-25-47558-T

The authors have extended their recent work [1] on diphenyl hexatriene (DPH) based singlet fission (SF) chromophores to now incorporate time-resolved EPR (trEPR) and optically detected magnetic resonance (ODMR) experiments. An additional methylated DPH trimer (Me-DPH₃) is also included in this paper. In the opinion of this reviewer, the key arguments about novelty are – 1.) Quintet mediated emission reported by low-temperature trEPR and ODMR (also at 300 K), 2.) a *platform* to selectively achieve the ⁵(TT) high-spin state from the ¹(TT) state formed by internal conversion from S* to ¹(TT).

While the former demonstration is highly interesting and novel, I find the evidence presented for the latter claim unclear. I elaborate my reasons below –

- a. Highly state-selective conversion of ¹(TT)₀ to ⁵(TT)₀ with absence of ³(TT) has been theoretically predicted and experimentally demonstrated in rigid, symmetric dimers. For example, see the theory work in ref. [2] and the experimental demonstration in Fig. 2 of ref. [3]. The predictability of their model provides a rational synthetic platform. Compared to this demonstration, Fig. 1 shows a large contribution from SF generated ³(TT) photoproducts. As pointed out by reviewer 3, aspects regarding the predictability of spin dynamics and a recipe for molecular design are lacking. I therefore think that [2,3] are quite relevant references in the context of the current work.
- b. Further to point a), the effect of methylation on the DPH unit has been loosely connected to planarity while changes observed in the linear absorption and time-resolved photoluminescence (TRPL) measurements have been interpreted as not being significant. Fig. S5 suggests that the net oscillator strength in the PL red shifts with temperature. Similarly, Fig. S12 shows that the net absorption oscillator strength blue shifts for the methylated trimer. Such trends in the oscillator strength are interesting and might hold vital clues about what the methylation is doing to the conformational landscape and the electronic delocalization, both key for predicting spin dynamics and providing a rational synthetic recipe. Along the same lines, PL lifetime in Fig. S12b likely shows lifetime and/or %Amplitude contribution changes that may not be insignificant. Unfortunately, no further analysis of the results in Fig. S12 is presented, which I think would really help the claim about a tunable platform for high-spin state dynamics. I can understand that modeling absorption spectra and electronic structure changes may be out of the current scope but a sound discussion about the shortcomings, as mentioned above, is necessary.

- c. I find the discussion regarding the complete re-interpretation of the ‘triplet plateau’, presented originally in [1], to be rather vague, as was also pointed out by reviewer 3. As I understand, previous work of the authors [1] reports a hypsochromic shift in the spectral feature for independent triplets compared to $^1(\text{TT})$, similarity with the triplet spectrum obtained from sensitization, and similarity of ‘triplet plateau’ lifetime with that of isolated triplets in order to justify the assignment of the plateau to isolated triplets arising from SF. In the current paper, the triplet plateau is instead assigned to come from the dynamics of the quintet. I think a proper discussion about the significant re-interpretation of the results in [1] will be quite useful. It also raises a fundamental question about the (in)validity of the above criteria to identify isolated triplet dynamics based on spectral similarities and lifetimes, which is rampant in the SF community.
- d. Triplet contributions are seen at 80K suggesting significant spin relaxation within the $^5(\text{TT})$ manifold and population flow into the $^3(\text{T}\dots\text{T})$ manifold already at 80K (Fig. 2). It is not clear to me what is the mechanism for selective depopulation of the $^3(\text{T}\dots\text{T})$ manifold into the $^5(\text{T}\dots\text{T})$ at room temperature resulting in ODMR showing no contributions from the triplet at all ? (Fig. 3c). The rate schematic in Fig. 5 does not consider this at all but I think this aspect is quite central to the results presented here.
- e. To support the claim that quintet/triplet ratio decreases with temperature, it will be useful to breakdown Figs. 3a and 3b into individual quintet, triplet contributions such as that in Fig. 2a. The caption of Fig. 3 says so but I do not see that in the figure. Inferring it directly from the figure is a stretch for untrained eyes.

In summary, I think the argument about Me-(DPH)₃ providing a tunable platform has several shortcomings that need to be addressed. The demonstration of quintet luminescence at 300K is interesting, although the mechanism for *only* quintet signal in ODMR at 300K is unclear.

Reference DOIs:

1. <https://doi.org/10.1021/jacs.4c10483>
2. <https://doi.org/10.1038/s41598-020-75459-x>
3. <https://doi.org/10.1038/s41467-023-36529-6>

Comments regarding manuscript NCOMMS-25-47558A-Z

I have carefully revisited my previous comments from other reviewers, and the corresponding changes made in the manuscript. I unfortunately do not find sufficient evidence for the two central claims about 1. “illustrating a framework for molecular design to *selectively* control spin-state formation” and 2. “a *design principle* for DPH-based materials as platforms”.

In response to reviewer comments, for example, comments 2 and 3 from reviewer 1, the authors now argue that both ^3TT as well as intersystem crossing (ISC) born triplets may be present and contribute to ODMR signatures. The argument is then made that the only reason that room temperature ODMR does not show triplet signatures is because ^5TT is long lived and the ^3TT and ISC born triples states are not. What is the mechanism behind this and therefore, whether this is a *tunable* property or not, is not clear from the evidence presented.

In another example, considering the response to my comment b, the authors cite (Kobori et al. J. Phys. Chem. B 2020, <https://pubs.acs.org/doi/10.1021/acs.jpccb.0c07984>) for connecting the effect of methylation to the appearance of ISC triplets below 80 K but not at 300K. The authors argue that the low-frequency THz motions, are enhanced by the methyl group and promote ^1TT to ^5TT mixing, thus competing against ISC at room temperature. It is proposed that at 80 K these motions “freeze out” to reduce the probability of ^1TT to ^5TT conversion and therefore the ISC signatures become more prominent at 80 K. This argument, on which the central claim about the tunability and predictability of the DPH platform rests, seems fairly speculative to me. 80 K is $\sim 55\text{ cm}^{-1}$. The low-frequency THz motions as discussed in Kobori et al. are only $\sim 72\text{ cm}^{-1}$ and therefore expected to play an active role even at 80 K. This is indeed reported by Figure 5 of Kobori et al. where these motions actively participate in ^1TT to ^5TT conversion which is evident at 77K. Thus, “freezing out” methyl motions does not seem like a tunable knob at all. Furthermore, ^5TT states dominate room temperature emission in all systems including $(\text{DPH})_3$ where there is no methyl group. Thus, even the promoting role of the methyl group in ^1TT to ^5TT conversion is not clear.

The above points also echo with the comments from reviewer 3 of the previous round, specifically points 2 and 3 – the unpredictability of spin dynamics and the study being focused on a methylated version of $(\text{DPH})_3$ studied earlier, respectively.